# Brain RFamide Neuropeptides in Stress-Related Psychopathologies

**DOI:** 10.3390/cells13131097

**Published:** 2024-06-25

**Authors:** Anita Kovács, Evelin Szabó, Kristóf László, Erika Kertes, Olga Zagorácz, Kitti Mintál, Attila Tóth, Rita Gálosi, Bea Berta, László Lénárd, Edina Hormay, Bettina László, Dóra Zelena, Zsuzsanna E. Tóth

**Affiliations:** 1Institute of Physiology, Medical School, Centre for Neuroscience, Szentágothai Research Centre, University of Pécs, H7624 Pécs, Hungary; anita.kovacs@aok.pte.hu (A.K.); szabo.evelin9812@gmail.com (E.S.); kristof.laszlo@aok.pte.hu (K.L.); erika.kertes@aok.pte.hu (E.K.); olga.zagoracz@aok.pte.hu (O.Z.); kitti.mintal@aok.pte.hu (K.M.); attila.toth@aok.pte.hu (A.T.); rita.galosi@aok.pte.hu (R.G.); beata.berta@aok.pte.hu (B.B.); laszlo.lenard@aok.pte.hu (L.L.); edina.hormay@aok.pte.hu (E.H.); bettina.csetenyi@aok.pte.hu (B.L.); 2Department of Anatomy, Histology and Embryology, Semmelweis University, H1094 Budapest, Hungary

**Keywords:** GPCR, anxiety, depression, HPA, coexpression

## Abstract

The RFamide peptide family is a group of proteins that share a common C-terminal arginine–phenylalanine–amide motif. To date, the family comprises five groups in mammals: neuropeptide FF, LPXRFamides/RFamide-related peptides, prolactin releasing peptide, QRFP, and kisspeptins. Different RFamide peptides have their own cognate receptors and are produced by different cell populations, although they all can also bind to neuropeptide FF receptors with different affinities. RFamide peptides function in the brain as neuropeptides regulating key aspects of homeostasis such as energy balance, reproduction, and cardiovascular function. Furthermore, they are involved in the organization of the stress response including modulation of pain. Considering the interaction between stress and various parameters of homeostasis, the role of RFamide peptides may be critical in the development of stress-related neuropathologies. This review will therefore focus on the role of RFamide peptides as possible key hubs in stress and stress-related psychopathologies. The neurotransmitter coexpression profile of RFamide-producing cells is also discussed, highlighting its potential functional significance. The development of novel pharmaceutical agents for the treatment of stress-related disorders is an ongoing need. Thus, the importance of RFamide research is underlined by the emergence of peptidergic and G-protein coupled receptor-based therapeutic targets in the pharmaceutical industry.

## 1. Introduction

### 1.1. Stress and Stress-Related Neuro-Circuitries

Homeostatic threats trigger the stress response, which is crucial for survival. However, maintaining homeostasis in situations of frequent or chronic stress requires continuous active effort. Too much stress can therefore have detrimental effects on health and can lead to the development of various diseases, such as cardiovascular disease, various types of cancer, endocrine disorders, and mental health problems [1].

The reaction to stress is organized via the central nervous system (CNS) and two output systems, the hypothalamic–pituitary–adrenal axis (HPA) and the sympathoadrenomedullary system (SAM, part of the autonomic nervous system), which are activated to generate appropriate physiological and behavioral responses (Figure 1). Stimulation of the HPA begins with activation of parvocellular neurons coexpressing corticotropin-releasing hormone (CRH) and arginine vasopressin (AVP) in the paraventricular nucleus (PVN) of the hypothalamus (HTH). CRH then stimulates the secretion of adrenocorticotropic hormone (ACTH) from the anterior pituitary, an effect potentiated via AVP. ACTH in turn induces the secretion of glucocorticoids (CORT), mainly cortisol in humans and corticosterone in rodents, from the adrenal cortex. Glucocorticoids, together with adrenaline released from the adrenal medulla during SAM activation, reach target organs via the circulation to modulate several physiological functions. A critical aspect of the HPA response to stress is the negative glucocorticoid feedback that affects pituitary and brain levels, which allows the HPA to return to its physiological state following acute activation (Figure 1) [2]. In contrast, increased sympathetic activity is counteracted through the activation of the parasympathetic part of the autonomic nervous system. This occurs mainly through the vagus efferents (cranial nerve X) from the dorsal motor nucleus (DMX) of the vagus nerve in the brainstem, which innervates most internal organs.

It is apparent that the structures that control the activity of the HPA and the autonomic nervous system are responsible for maintaining stress sensitivity and keeping the stress response within a healthy range. These include many brainstem, hypothalamic, limbic, and cortical areas (Figure 1). Information about homeostatic stress is transmitted to higher centers via the catecholamine cell groups of the brainstem, the C1/C2 adrenergic and A1/A2 noradrenaline (NA) cell groups in the medulla oblongata, and the NA cells in the locus coeruleus (LC, cell group A6). Ascending catecholaminergic fibers stimulate the HPA via directly stimulating CRH neurons of the PVN. Simultaneously, they stimulate cortical and limbic areas involved in emotional and cognitive control, emotional learning, and memory formation, such as the prefrontal cortex, amygdala (AMY), and hippocampus (HC) (Figure 1) [3]. These areas contain large amounts of glucocorticoid receptors and influence the activity of the HPA mainly indirectly through the bed nucleus of the stria terminalis (BNST), the lateral septum, the subparaventricular region, and various hypothalamic nuclei [4,5,6]. On the other hand, the AMY, the HC, and the prefrontal cortex, together with the cingulate cortex and the insular cortex, are also part of the central autonomic network that exerts the highest control over the autonomic outflow from the hypothalamus and the brainstem (Figure 1). As an integrative center of autonomic and neuroendocrine responses, the HTH controls the lower autonomic centers, such as the brainstem premotor neurons in the periaqueductal gray (PAG), parabrachial nucleus (PBN), medullary raphe, and sympathoexcitatory catecholamine cell groups and innervates the sympathetic preganglionic neurons in the spinal cord. The PVN plays a pivotal role in this system, as it is the center of the HPA and is also directly connected to both brainstem and spinal autonomic neurons [3,7,8].

HPA hyperactivity and autonomic nervous system dysfunction as well as structural and functional impairment of the limbic–cortical circuit underline stress-related psychopathologies such as depression, anxiety disorders, post-traumatic stress disorder (PTSD), and eating disorders [1,9,10,11]. These diseases are characterized by a disruption in the signal transmission within the affected brain regions. This is typically manifested through an imbalance in the ratio of excitatory to inhibitory amino acid neurotransmitters and a disturbance in the function of neuromodulatory monoaminergic systems [12,13]. In fact, serotonin and NA reuptake inhibitors are the first-line treatment for stress-related psychiatric disorders. It is therefore unfortunate that they are ineffective in 30% of cases [14]. Consequently, there is an urgent need for identification of novel pharmacotherapeutic targets.

### 1.2. RFamide Peptides as Promising CNS Targets for the Treatment of Stress-Related Disorders

Peptidergic neuromodulators and their G-protein coupled receptors (GPCRs) have promising therapeutic potential and are attracting increasing interest [15,16]. Neuropeptides are coexpressed with classic neurotransmitters, monoamines, and other neuropeptides. They are released under challenged conditions and help to fine-tune cellular activity [17,18,19]. The expression of neuropeptides is usually restricted to a small population of neurons, and they bind to their receptors with high affinity and specificity. Dysregulation of several neuropeptides is seen in neuropsychiatric disorders [17,20].

RFamide peptides in the brain may represent key targets for the treatment of stress-related disorders, given their involvement in the regulation of stress response and numerous other aspects of homeostatic regulation [21]. These include, for example, the regulation of the energy balance, the reproductive axis, and the modulation of pain perception. In addition, several RFamide peptides modulate memory and learning processes and activity. (For a summary and references to these effects of RFamide peptides, see Table 1.) Most of these functions are adversely affected by chronic stress. Due to their multifunctionality, RFamide peptides may therefore provide a link between stress regulation and other homeostatic functions.

This review focuses on the potential role of RFamide peptides in the central nervous system, as a possible key hub in the regulation of the stress response and stress-related mental disorders. As described in Section 1.1, stress-related brain areas are distributed in a hierarchical manner across the brain. Therefore, an overview of the distribution of RFamide peptides and their receptors is provided, which offers insight into their function. Furthermore, the coexpression profiles of neurotransmitters in cells expressing RFamide peptides and the stress-related roles of different groups of RFamide peptides are discussed.

## 2. The RFamide Peptide Family and Their Receptor Promiscuity

The peptides belonging to the RFamide family have an arginine (R)-phenylalanine (F)-amide motif at the C-terminal of their amino-acid sequence, hence their name: RFamide peptides. To date, five groups of the RFamide peptides have been discovered in mammals: the neuropeptide FF (NPFF) group, the LPXRFamide peptide/mammalian RFamide-related peptides (RFRPs) group, the prolactin-releasing peptide (PrRP) group, the pyroglutamylated RFamide peptide (QRFP) group, and the kisspeptin (KP) group (Figure 2) [96,97,98].

The NPFF and LPXRFamide/RFRP peptides share common receptors, which are the NPFF receptor 1 (NPFFR1 or GPR147) and the NPFF receptor 2 (NPFFR2 or GPR74), showing 50% homology with each other [56,99,100,101]. The two families differ in their affinity for these receptors, with the NPFF group preferring the NPFFR2 and the LPXRFamide/RFRP group preferring the NPFFR1 (Figure 2) [21,102,103,104]. PrRPs, QRFPs, and KPs have their own cognate receptors showing high affinity and selectivity for their ligands. These are the PrRP receptor (PrRPR, alias human PRLHR, GPR10, hGR3, rat UHR-1) for the PrRP family [56,105], the QRFP receptor (QRFPR or GPR103, previously referred to as AQ27 or SP9155) for the QRFP family [106,107,108], and the KP-1 receptor (Kiss1R or GPR54) for the KP family (Figure 2) [109,110,111].

However, in vitro pharmacological studies showed that all RFamide peptides were able to bind to and activate both types of NPFFRs and reduce the basal nociceptive threshold in mice, which could be prevented through administration of RF9, a putative NPFFR1/2 antagonist (Figure 2) [63,112]. Indeed, NPFFR2 is known to induce heterologous desensitization of the mu-opiate receptor involved in the central modulation of pain [113]. Yet, several studies have argued against RF9 being a selective NPFFR2 antagonist [114,115,116]. In fact, in vivo data did not support the involvement of NPFFRs in the mechanism of action of QRFP peptides [75]. Nevertheless, the biological activity of KPs on NPFFRs was confirmed by two independent groups, one using acute brain slices from Kiss1R knockout (KO) mice [117,118]. Likewise, PrRP has high affinity and efficacy at NPFFR2s [63,119], and the central effects of PrRP on the cardiovascular system are indeed mediated via NPFFR2s [120]. Moreover, PrRP also reduced cortical excitability in rats via activating NPFFR2s but not NPFFR1s (Figure 2) [121].

The cross-reactivity of RFamide peptides on NPFFRs may underlie the mechanism of their diverse biological action.

## 3. Discovery of the RFamide Peptides

### 3.1. NPFF Peptides

The NPFF peptide group contains small peptides encoded in the Npff (farp1) gene [122]. The Npff gene encodes two precursor polypeptides, pro-NPFFA and pro-NPFFB [56]. Processing of pro-NPFFA yields peptides of the NPFF group which share the common feature of having a C-terminal proline(P)–glutamine(Q)–RFamide sequence, whereas processing of pro-NPFFB yields RFRP peptides [56,102,122,123]. Derivates of pro-NPFFA include NPFF, neuropeptide AF (NPAF), and neuropeptide SF (NPSF) [123]. NPFF was the first identified mammalian neuropeptide belonging to the RFamide peptide family and was characterized as a pain modulator (Table 1). Both NPFF and NPAF were isolated from bovine brain extracts via affinity column chromatography, based on a cross-reaction with an antiserum produced against the molluscan RFamide sequence [67]. All peptides were later found in rats, mice, and humans [70,123]. The octapeptides NPFF and NPSF were named after their phenylalanine (F)–8-PQRFamide and serine (S)–8-PQRFamide sequences, respectively. Similarly, NPAF has an alanine (A)-18-PQRFamide sequence, which is reflected in its name. 

The peptides derived from the pro-NPFFA and pro-NPFFB precursor molecules are therefore related and bind to identical receptors, which explains why the NPFFR1/2s are the cognate receptors for both the NPFF and LPXRFamide/RFRP families of peptides [56,99,100,101].

### 3.2. LPXRFamide/RFRP Peptides

LPXRFamide peptides with a C-terminal leucine(L)–Proline(P)–X-RFamide (X= variable) motif are important molecules in reproduction, as they are potent inhibitors of the hypothalamic–pituitary–gonad axis (Table 1) [54]. The first RFamide peptide identified in vertebrates was the avian gonadotropin-inhibitory hormone (GnIH) in 1983 [124]. 

This review addresses the topic of RFRPs, since these peptides represent the mammalian orthologues of GnIH. The family is therefore referred to as RFRPs throughout the text. Although the preproRFRP (pro-NPFFB) precursor protein contains three RFRP sequences (RFRP-1-3) in humans, only RFRP-1 and RFRP-3 have been isolated from human HTH [125]. The RFRP-2 sequence is absent from the rodent preprotein [56] and was later excluded from the entire family because it did not contain the LPXRFamide sequence [126]. The family-specific C-terminal sequence for the RFRP-1 peptide is LPLRRFamide [127] and that for RFRP-3, known also as neuropeptide VF (NPVF) in humans, is LPQRFamide [126]. RFRP-3 suppresses plasma luteinizing hormone (LH) levels upon intracerebroventricular (ICV) administration [52,54,55] and inhibits the activity of gonadotropin-releasing hormone (GnRH) neurons that drive LH release from pituitary gonadotrophs [51]. However, in intact diestrus mice, RFRP-3 did not influence blood LH levels. Furthermore, in intact and castrated male mice, as well as in photoperiodic male hamsters kept under a long-day photoperiod, RFRP-3 stimulated the release of LH, indicating that its effect is highly dependent on the hormonal status of the subject (Table 1) [54]. 

### 3.3. PrRPs

PrRP was named in the hope of discovering a hypophysiotropic factor that positively regulated the release of prolactin [128]. It was identified from bovine, rat, and human brain tissue samples as a ligand of the orphan receptor GPR10 (hGR3 or UHR1) isolated from the human pituitary gland. Collective morphological and experimental evidence based on rodent and human studies later disproved that stimulation of prolactin release was the physiological function of PrRP [129,130,131,132,133]. In contrast, PrRP was found to be important in the regulation of energy homeostasis. In rodents, it reduces nocturnal food intake and increases energy expenditure (core body temperature) via mediating the actions of leptin and cholecystokinin (Table 1) [28,29,30,43]. The PrRP group has two members, the longer PrRP31 (31 amino acids) molecule and the shorter PrRP20 molecule containing C-terminal 20 residues of PrRP31 [128]. The structure of the promoter region of the rat PrRP gene suggest a possible tissue-specific expression of the PrRP isoforms [134], but the biological significance of them has not been revealed yet. 

### 3.4. QRFPs

QRFPs have two isoforms, the N-terminally truncated QRFP-26 (26RFa) and QRFP-43 (43RFa), consisting of 26 and 43 amino acids, respectively. The family was identified simultaneously by three independent teams and 26RFa was first isolated from frog brain [44,106,107]. Both isoforms were identified in rat HTH [46] and were isolated from a culture medium of Chinese hamster ovary cells that expressed the human peptide precursor [135]. Both forms exert similar physiological effects, although some studies have suggested that the elongated form of the peptide is more potent. The physiological functions of QRFP peptides are diverse (Table 1), and increasing the food intake is their most studied and consistent effect [32,44,45,46,47]. The cognate receptor of QRFP peptides is the glutamine RFamide peptide receptor (QRFPR), formerly the orphan receptor GPR103, which exists in two isoforms in rodents [46,107,108].

### 3.5. Kisspeptins

KPs are a group of proteins encoded in the Kiss1 (in animals)/KISS1 (in humans) genes [136,137] that act as endogenous ligands of the same orphan receptor, GPR54, known today as Kiss1R/KISS1R in animals and humans, respectively [110,111,138]. The name KP comes from the name of the Pennsylvania chocolate “Hershey’s Kisses”, as it was discovered in Pennsylvania, where an antimetastatic effect of the KISS1 gene product metastin was first demonstrated in 1996 [136]. Metastin was subsequently renamed KP-54 (corresponding to KP-52 in rodents) upon its identification as the most potent 54-amino-acids-long fragment of the 145-amino-acids-long KP precursor molecule, exhibiting the longest half-life. Other biologically active fragments are peptides of 14, 13, or 10 amino acids having a common C-terminal amidation site that enables strong binding with their receptor [110,138,139]. KPs turned out to be an essential element in the central regulation of reproduction [140]. Similarly to RFRPs, they are involved in the regulation of the hypothalamic–pituitary–gonadal axis (HPG). Kisspeptins (KPs) play a pivotal role in the regulation of gonadotropin-releasing hormone (GnRH) release. Consequently, they determine fertility, the onset of puberty, and reproductive behavior (Table 1) [61,62,141]. Another crucial aspect of the KP function is the integration of metabolic and reproductive regulation, which has been demonstrated to be a key factor in fertility [142,143,144].

## 4. Distribution of RFamide Peptides and Their Receptors in the CNS

### 4.1. NPFF and NPAF

#### 4.1.1. Cell Bodies

In rats, NPFF-immunoreactive (IR) neurons are mainly confined to the medial HTH, medulla oblongata, and the spinal cord [145,146], and they were recently found in the cerebellum of mice [147]. In the HTH, the neurons are distributed between the ventromedial (VMN) and dorsomedial (DMN) regions, the hypothalamic nuclei, and the periventricular hypothalamic nucleus (PeN). NPFF neurons extend into the most caudal parts of the DMN, the tuberal magnocellular nucleus, and the arcuate nucleus (ARC) [145]. In the brainstem and spinal cord, NPFF neurons are located in the rostral nucleus of the solitary tract (NTS) (containing autonomic and sensory regions) and in the superficial layers of the dorsal horn, as well as around the central canal [146]. In addition, Goncharuk et al. [148] demonstrated high density of NPFF-IR cells in the rat supraoptic nucleus (SON), a center for osmotic regulation that produces AVP and oxytocin. These two hormones play a role in the regulation of the stress response and are involved in the pathomechanism of human depression [149]. Moderate density of NPFF-IR cells has been demonstrated in the anterior amygdaloid area, the horizontal limb of the diagonal band, the medial forebrain bundle, and in the center of the stress-regulatory HPA, the PVN. They also found small numbers of scattered cells in the basal nucleus, the BNST, a node for sustained anxiety-related responses [150], the lateral hypothalamic area (LHA), the lateral tuberal nucleus, the perifornical hypothalamic nucleus, the posterior hypothalamic area, and the zona incerta of rats. 

The distribution of NPFF-IR neurons is similar in the forebrains of humans and rats, with the difference that in humans, NPFF-IR neurons have been detected in the suprachiasmatic nucleus, the circadian master-clock, deeply influencing our mood [151], whereas no NPFF-IR cells were found in the SON [148]. NPFF is also present in human cerebrospinal fluid [152], and its concentration in human serum shows an ultradian but not diurnal rhythm [153,154].

Since NPFF and NPAF are closely related, antibody specificity is an important issue in the immunohistochemical detection of these peptides. In this regard, it is important to note that Aarnisalo et al. [155] reported separate localization of NPAF and NPFF immunoreactivity in rat brains and spinal cords and described a limited distribution of NPAF-IR neurons restricted to the magnocellular cells of the PVN and SON.

#### 4.1.2. Fibers

NPFF-IR fibers primarily target limbic (e.g., BNST, septal nuclei, accumbens nucleus, nucleus of the diagonal band, medial AMY, medial mammillary nucleus, anterior thalamus) and hypothalamic areas (e.g., preoptic region, anterior HTH, PeN, suprachiasmatic nucleus, PVN, SON, DMN, VMN, ARC, tuberal magnocellular nucleus) in the rat forebrain [145,148]. Interestingly, a few NPFF-positive fibers pass the median eminence to innervate the posterior pituitary gland. The paraventricular nucleus of the thalamus [156], considered a main node of the brain’s anxiety network, also receives NPFF innervation, albeit only to a moderate extent. Fibers descend to the autonomic and pain-related centers of the brainstem (e.g., lateral PBN, reticular formation, NTS, dorsal tegmental nuclei, caudal parts of the spinal trigeminal nucleus), but the density of axons around the PAG, a center for pain modulation, is low [145,148,157]. Distribution of NPFF-positive fibers in the forebrain is similar in humans and rats, with poor innervation of the perifornical hypothalamic nucleus in rats but dense innervation in humans [148]. Further investigation revealed the presence of NPFF fibers in the sensory, autonomic, and motor regions of the rat spinal cord (laminae I-IV and X, IML, sacral parasympathetic nucleus, ventral horn) [146].

In contrast to NPFF, the presence of NPAF-IR fibers was observed in the median eminence and the posterior pituitary, as well as in the commissural (autonomic) part of the NTS, but not in the spinal cord. No colocalization was seen with NPFF-positive fibers [155].

### 4.2. RFRPs 

#### 4.2.1. Cell Bodies

RFRP-1 and RFRP-3 immunoreactivities overlapped without sex difference in male rats [158]. Nevertheless, RFRP-producing neurons were detected in a restricted area of the HTH of rodents, in the DMN, PeN, and in an area between the VMN and DMN extending to the LHA [56,158,159,160]. To date, these are the only areas where RFRP positive neuronal cell bodies have been localized via in situ hybridization (ISH) and immunohistochemistry (IHC) in gonad-intact rodents [161]. However, in ovariectomized (OVX) estrogen-primed rats, RFRP-IR cells appeared in the ARC as well [52]. RFRP-producing neurons were also detected also in tissue samples from postmortem human subjects using IHC in the DMN [125].

#### 4.2.2. Fibers

Given the relatively limited number of neurons expressing RFRPs, the distribution of RFRP-IR fibers in the brain is surprisingly extensive [158]. According to the function of RFRP as a regulator of GnRH cells, these include the preoptic area (POA) and the ARC in rodents [158,159], where GnRH and KP neurons are located [55,162]. Indeed, RFRP fibers establish close contact with GnRH neuronal cell bodies in rats [55] as well as in humans [125]. The fibers also reach the internal layer of the median eminence both in rodents and humans [53,159,160], where GnRH axons terminate, and release their hormones into the pituitary portal circulation. Thus, RFRPs may directly modulate the function of GnRH neurons, acting on both the cell bodies and axon terminals. Furthermore, RFRPs can regulate GnRH cells indirectly via acting on KP neurons [161]. Indeed, ICV applied RFRP-3 inhibited KP protein expression and activity [52,53].

Other target areas of RFRP neurons suggest additional functions of RFRP, such as processing emotional and stress information and regulation of energy balance. Thus, similarly to NPFF neurons, RFRP neurons also innervate limbic structures and autonomic stress centers, such as the BNST, lateral septal nuclei, AMY, nuclei of the diagonal band, PVN, DMN, VMN, paraventricular, lateral habenular and thalamic reuniens nuclei, dorsal raphe nucleus, Edinger–Westphal nucleus, PBN, PAG, LC, lateral reticular nucleus, NTS, and spinal trigeminal nucleus, among other regions [55,158,159,160]. In the ovine HTH [163], RFRP-3 axons come into contact with oxytocin, neuropeptide Y (NPY), and pro-opio-melanocortin (POMC) neurons, which play a critical role in energy balance regulation [164], orexin, and melanin-concentrating hormone (MCH) neurons, which are central nodes in the integrative regulation of sleep–wake states, energy homeostasis, reward system, cognition, and mood [165,166], as well as CRH neurons in the PVN, constituting the apex of the HPA.

#### 4.2.3. Distribution of NPFF Receptors

Both NPFFR1, exhibiting a higher affinity for RFRPs than for members of the NPFF peptide family, and NPFFR2, showing higher affinity for NPFF peptides (Figure 2) [21,102,103,104], are widely expressed in the CNS. However, substantial species differences exist among mammals in the distribution of these receptors [167]. Both receptors utilize Gi/Go coupled signal transduction pathways and can also bind to Gs protein, but the exact pathways are still poorly defined [100,168].

Reverse transcriptase polymerase chain reaction (RT-PCR) revealed the highest levels of NPFFR1 in the spinal cord in humans, whereas in rats, these were found in the hypothalamus [101]. Strong NPFFR1 expression has been observed in the human HC, AMY, and thalamus, while somewhat lower expression was detected in the HTH and the cerebellum. In rats, high levels of NPFFR1 were present in the olfactory bulb, AMY, accumbens nucleus, and substantia nigra, while moderate levels were found in the cortex, choroid plexus, HC, medulla, and spinal cord (Figure 3). 

Human NPFFR2 expression is generally weak in the CNS, and a moderate level of expression has been measured in the AMY. However, CNS data were expressed relative to the very high amount of RNA measured in the placenta, which influenced the CNS outcome. In rats, however, NPFFR2 was very strongly expressed in the HTH, substantia nigra, medulla, and spinal cord, moderately expressed in the AMY, choroid plexus, and retina, and weakly expressed in the thalamus (Figure 3) [101]. The central distribution of NPFFRs in mice was analyzed at cellular resolution via radioactive ISH [169] and in rats via RNAscope ISH [170]. The results slightly differed from those obtained with RT-PCR. Confirming previous data, the NPFFR1 signal was strongest in the HTH, mainly in the PVN, PeN, ventromedial HTH, and AMY (medial and central parts being key structures in emotional processing and fear response) [171,172] of rats. The signal was also strong in the BNST, lateral septum, ARC, superior colliculus, median raphe, and area postrema (AP) (Figure 3). In the rostral periventricular region of the third ventricle (AVPV), NPFFR1 expression was strong in diestrus females, but weak in males [170]. 

NPFFR2-producing cells appeared mostly in the lateral lemniscus and the principal and spinal trigeminal nuclei; moderate expression of NPFFR2 was seen in the zona incerta, HTH (subparaventricular region, POA, PVN, DMN), thalamus, NTS, and DMX in rats [170], while in mice, the olfactory bulb, ARC, thalamus, dorsal tegmentum, and NTS showed the strongest labeling (Figure 3) [169]. 

These overall findings were in harmony with early data received via receptor autoradiography; in broad terms, the distribution of NPFFR1 refers more to involvement in neuroendocrine functions, whereas the distribution of NPFFR2 refers more to involvement in somatosensory pathways [167]. 

#### 4.2.4. The Chemical Nature of NPFFR-Bearing Cells

In the AVPV, where NPFFR1 expression was sex-dependent, 52% of the KP neurons expressed NPFFR1. In rats, the majority of CRH neurons in the PVN (70%) and dopaminergic neurons in the PeN (79%) and the ARC (43%), as well as many orexigenic NPY (20%) and a few anorexigenic POMC (6%) neurons in the ARC expressed NPFFR1. [170]. The chemical nature of NPFFR2-expressing cells in the ARC was also heterogeneous. In mice, the majority of neurons expressing NPFFR2 mRNA were gamma-amonibutyric acid (GABA, a main inhibitory neurotransmitter)-producing cells (64%), but some (21%) were glutamate (a main excitatory neurotransmitter)-expressing neurons. NPFFR2 positivity was abundant in both NPY (64%) and POMC (40%) cells, but NPFFR2-expressing cells were also detected in the somatostatin- (28%), dopamine- (26%), nociception- (16%), and NPFFR1- (16%) producing neuronal populations [173,174]. Human dual ISH data demonstrated that, similarly to mice, NPFFR2 mRNA was predominantly expressed in NPY- and GABA-producing neurons of the ARC, whereas in contrast to mice, POMC neurons did not show any NPFFR2 signal [175]. These data suggest that NPFFRs are central mediators of several aspects of the regulation of homeostasis, including the stress response. In addition, NPFFR1 and 2 immunoreactivity coexisted in GAD-67 and TH immunopositive neurons in the ventral tegmental area (VTA) [176], which may have provided a morphological basis for the effects of NPFF on the mesolimbic pathway (Table 1). 

### 4.3. PrRP 

#### 4.3.1. Cell Bodies

PrRP neurons are found in only three areas of rat and mouse brains: the caudal-ventral part of the DMN, the caudal NTS, and the caudal ventrolateral medulla oblongata [130,177,178,179,180,181]. The presence of PrRP mRNA has also been confirmed in the human medulla oblongata [182] and pituitary [183]. The expression of PrRP is strongest in the NTS and weakest in the DMN [56,184,185]. Furthermore, the expression of PrRP in the medulla oblongata is subject to gonadal regulation, with the highest expression observed in the proestrus phase in female rats [185,186,187]. Accordingly, medullary but not hypothalamic PrRP cell groups showed estrogen receptor alpha immunopositivity in female rats [185,187].

#### 4.3.2. Fibers

PrRP fibers primarily target the HTH [181], which is probably also the case in humans [188] and suggests functions in homeostatic regulation. The most densely innervated areas are the PVN, the DMN, the perifornical area, and the LHA [130,177,189,190]. Several hypothalamic areas receive moderate PrRP innervation, such as the anteroventral periventricular area (AV3V), the magnocellular nuclei, and the PeN, including the ependymal layer. Practically, no PrRP fibers are detected in the median eminence [130,177,189] confirming that PrRP does not function as a neurohormone. Non-hypothalamic PrRP efferents target stress-related emotional and autonomic centers: the BNST, the septal nuclei, the central AMY, the paratenial thalamic nucleus, the AP, and the NTS [19,130,177]. PrRP axon terminals form close contacts with CRH, oxytocin, and somatostatin neurons in the PVN [130,177].

#### 4.3.3. Distribution of PrRP Receptors

It was revealed that in vertebrates, the PrRPR family consisted of two subtypes, named PrRPR1 and PrRPR2 [191]. Since, however, PrRPR2 has not been identified in the mammalian lineage, the distinctive nomenclature is not widely used in the literature, which focuses mainly on mammals. The PrRPR1, referred to as PrRPR protein, shares high sequence identity with the human NPY receptor type 2. Importantly, when PrRP31 and NPY were together added to human embryonic kidney cells at concentrations corresponding to their inhibitory constant values, NPY effectively inhibited the intracellular Ca^2+^ response to PrRP31 [191]. Although it is an interesting question whether NPY can act as a competitive antagonist of PrRP in vivo, this has not been investigated further. Nevertheless, the intracellular signaling pathways of PrRPR are diverse. As with many other members of the RFamide family, PrRPR acts through Gi/Go proteins [192]. However, depending on the cellular system under investigation, Gq and Gs pathways have also been proposed to be involved in the signal transduction mechanisms of PrRPR [193].

Our detailed knowledge of the expression pattern of PrRPR comes mainly from studies in rats. Within the brain, PrRPR mRNA expression is the strongest in the reticular nucleus of the thalamus, where paradoxically, no PrRP axons are found. High or moderate PrRPR expression has been detected in the BNST, HTH (preoptic nuclei, AV3V, DMN, PeN, PVN), AMY, LC, NTS, and AP (Figure 3). Several other brain areas, such as the perifornical area, the LHA, and the PBN show lower levels of PrRPR expression [19,179,190,194,195].

Human data are scarce and lack morphological details. Tissue homogenates of the large brain regions all contained PrRPR mRNA, except for the midbrain [196]. Expression of PrRPR mRNA was detected in the dorsal hypothalamic area and LHA in postmortem human hypothalamic samples via RT-PCR [190].

#### 4.3.4. The Chemical Nature of PrRPR-Bearing Cells

PrRPR-expressing cells in the reticular nucleus of the rat thalamus are GABAergic neurons [197]. This nucleus regulates sleep–wake states and has recently been implicated in the regulation of depressive-like behaviors induced via chronic stress/pain [198].

The majority of the CRH neurons in the BNST and several of them in the central AMY express PrRPR, confirming the role of PrRP in stress. PrRPR is also intensely coexpressed with pro-enkephalin mRNA in the PBN and central AMY, a center of the fear response [172,195]. Furthermore, oxytocin neurons playing a leading role in environment-dependent stress responses, and AVP neurons in the PVN, SON, and BNST exhibit PrRPR immunoreactivity [199,200].

### 4.4. QRFP

#### 4.4.1. Cell Bodies and Fibers

QRFP expression in the CNS has been confirmed both in rodents and humans [46,107,135]. In mice, QRFP mRNA-expressing cells were found via ISH in the HTH; namely in the PeN, LHA, and tuber cinereum areas [46]. In humans, QRFP mRNA expression was the strongest in the retina, the cerebellum, and the vestibular nuclei in a human autopsy tissue panel [107], while ISH and IHC revealed that QRFP neurons were localized in the PVN, the PeN, the VMN and in the dorsal and lateral horns of the spinal cord [135].

The distribution of the QRFP immunoreactive fibers remains to be elucidated.

#### 4.4.2. Distribution of QRFP Receptors

QRFPs were discovered as the endogenous ligands of the previously orphan receptor GPR103 [106,107,108]. While humans possess only one QRFP receptor isoform (GPR103), two distinct homologues have been identified in the mouse and rat genomes (termed GPR103a and GPR103b, or QRFPR1 and QRFPR2, respectively) [31,46]. Rat QRFPR1 and QRFPR2 share high amino acid identity with their mouse and human homologues and with each other. Moreover, QRFPRs share nearly 50% sequence identities with NPFFR1, NPFFR2, and orexin receptors. They are also related to NPY receptor 2, galanin receptor 1, and cholecystokinin receptors [101,107,108].

Studies regarding QRFPR mRNA expression in rodents suggest a broad receptor distribution within the CNS, with the highest expression in olfactory-related regions, such as the olfactory bulb, piriform cortex, and cortical AMY, and in other limbic structures like the amygdalohippocampal area, presubiculum, subiculum, BNST, and septum (Figure 3). Strong ISH signals have also been reported in the cingulate cortex, certain thalamic nuclei, zona incerta, and the HTH, with the highest signal density in the retrochiasmatic nucleus, PeN, POA, VMN tuberal nucleus, and ARC (Figure 3). In the brainstem, several nuclei involved in the regulation of vigilance states and alertness exhibit high levels of QRFPR expression: the interpeduncular nucleus, LC, medial PBN, pontine raphe, and dorsal raphe nuclei. A few cells with strong signal intensity have been observed in the ambiguous nucleus innervating the heart (Figure 3). Certain sensory areas also show high QRFPR expression, especially the vestibular nuclei and the dorsal horn of the spinal cord [31,96,201]. Bruzzone et al. [201] reported that QRFP binding sites in the rat CNS had a much wider distribution than areas of QRFPR mRNA expression. Such findings suggest that the neuropeptide QRFP might be involved in the activation of receptors other than QRFPR, thus implicating multiple pathways of action. QRFPR expression has also been demonstrated in the human brain, primarily in the cerebral cortex, HTH, thalamus, vestibular nucleus, and trigeminal ganglion [107,108]. Moderate expression also occurs in the AMY, caudate nucleus, HC, and the VTA area [107].

In cultured rat anterior pituitary cells preincubated with the adenylyl cyclase stimulator forskolin, QRFP provoked a dose-dependent increase in cAMP production, suggesting that the QRFP primarily stimulated the adenylyl cyclase enzyme through a stimulatory Gα subunit of the QRFPR [44]. This proposal was confirmed in adrenocortical and hypothalamic cells [202,203]. QRFPR also couples to the Gq protein, leading to activation of the mitogen-activated protein kinase (MAPK)/extracellular signal-regulated kinase ½ (ERK½) pathway. In transiently transfected HEK293 cells, QRFPRs can form functional heterodimers with orexin receptors, and binding of QRFP, orexin-A, or orexin-B ligands to these heterodimers induces ERK 1/2 phosphorylation [204]. It thus appears that QRFPRs, like most GPCRs, form dimers and display multiple signaling pathways that might account for the versatile activities of QRFP [205,206]. Nevertheless, in contrast to NPFFRs and PrRPR, the evidence does not indicate the involvement of a Gi/Go-mediated signaling pathway in the mechanism of action of QRFPR.

#### 4.4.3. The Chemical Nature of QRFPR-Bearing Cells

Consistent with the orexigenic effect of QRFP, coexpression of QRFPR and NPY was demonstrated in 12% of NPY neurons in the rat ARC [207].

### 4.5. Kisspeptins

#### 4.5.1. Cell Bodies

KPs (mRNA and protein) are expressed mainly in the rostral periventricular region of the third ventricle (AVPV-PeN) and in the ARC in mammals, and in equivalent regions in humans [138,162,208,209]. Few KP-IR neurons were detected in the DMN in male and female mice [210] and these were not confirmed via ISH performed in male mice [138]. Outside the HTH, KP-producing neurons were detected in the medial AMY and the BNST [211,212,213]. The distribution of KP in the mouse brain was recently mapped using CRE-activated tdTomato KP-reporter mice [214]. In addition to the well-known classical hypothalamic areas, small numbers of tdTomato-expressing, presumably KP-producing cells were found in the lateral septum, anterodorsal preoptic nucleus, AMY, medial preoptic nucleus, anterior hypothalamic area, DMN, VMN, PAG, and the mammillary nucleus. However, double-labeling in AVPV-PeN revealed co-expression of tdTomato and KP in only about two-thirds of tdTomato-positive neurons in males and about 20% in females. Ectopic dtTomato expression can occur for a number of reasons, including transient expression of the Kiss1 gene during development or levels of KP that are too low to be detectable with IHC [214].

The quantity of KP expressing neurons shows in the HTH shows a great difference between sexes [210,215], and the expression of KP in female rats and mice is negatively and positively regulated via estrogen in the ARC and the AVPV-PeN, respectively [212,215,216]. Consistently, human males lack KP-IR neurons in the AVPV-PeN region and have very low numbers of KP neurons in the infundibular nucleus (corresponding to the rodent ARC) compared with females [217]. Furthermore, investigation of tissue samples from pre- and postmenopausal women as well as from control and OVX monkeys suggested that estrogen negatively regulated KP mRNA expression in the infundibular nucleus of humans and monkeys [208], which was similar to rodent ARC. 

#### 4.5.2. Fibers

Distribution of KP-IR fibers was investigated in hypothalamic sections from human females and KP-positive neurons were detected nearby as well as in proximity to the third ventricle, including the organum vasculosum of the lamina terminalis, PVN, PeN, and DMN. An especially dense fiber network surrounded the capillary plexus of portal vessels in the infundibular stalk. Scattered fibers were described in the septal nuclei, LHA, and VMN [217]. Similar results were obtained in female mice, where KP-IR fibers were additionally found in the SON, BNST, paraventricular nucleus of the thalamus, medial AMY, PAG, and the vicinity of the LC [209].

#### 4.5.3. Distribution of Kiss1R

Kiss1R was identified in 2001, when the orhan receptor GPR54 was identified as the receptor for metastin (KP-54) [109,110,111]. The human orthologue of the receptor is KISS1R, previously known as AXOR12 or hOT7T175. Based on sequence similarities, Kiss1R is related to the galanin receptors [218]. It is a Gq-coupled receptor, whose activation leads to the release of intracellular Ca^2+^ and phosphorylation of various MAPKs such as ERK1/2 and possibly p38 [219]. Phosphorylation of ERK1/2 in response to Kiss1R activation can also occur Gq-independently via beta-arrestin1/2, through a signaling pathway that has a crucial role in KP-induced GnRH release in mice [220,221], as well as via focal adhesion kinase and steroid receptor coactivator signaling, a pathway involved in the regulation of motility of endometrial cancer cells [222]. Additional signaling pathways of Kiss1R may include the release of arachidonic acid [139]. In the prevention of metastasis, KPs have been reported to inhibit certain chemokine signaling routes, like that of the C-X-C chemokine receptor type 4 receptor (CXCR4) [223].

The expression of KISS1R mRNA in the human brain was measured via RT-PCR and found to be expressed in different regions, including the HTH (Figure 3) [109]. Accurate mapping of Kiss1R mRNA distribution in the brain was performed using transgenic mice and via ISH in rats [224,225,226]. High level of expression was detected in the olfactory bulb, medial septum, diagonal band of Broca, and HTH (Figure 3), [224,225,226]. Moderate Kiss1R was observed in the PVN and ARC of rats [224], while weak Kiss1R signal was seen in the supramammillary nuclei, dorsal raphe, and PAG in both rodent species [224,225]. In mice, Kiss1R expression was detected in areas where it was not observed in rats, such as the HC, thalamus, and several brainstem nuclei, including certain tegmental nuclei and sensory nuclei (dorsal cochlear nucleus, superior colliculus, cuneate nucleus) [225]. On the other hand, Kiss1R signal was strong in the amygdala of rats, a region where it was not reported in mice (Figure 3) [218].

#### 4.5.4. The Chemical Nature of the Kiss1R-Bearing Cells

In the forebrain, Kiss1R is expressed mainly in the GnRH neurons [224,225,226]. Moderate Kiss1R was observed within a small population of the oxytocin neurons in the rat PVN [224]. Although Kiss1Rs were not detected in the ARC of mice, it is important to highlight that in the ARC of ewes, Kiss1R expression was not observed in KP-positive neurons, but rather in GABA- or estrogen receptor alpha-expressing neurons [227]. In the rat ARC, the majority of neurons expressing Kiss1R (63%) were POMC neurons, while a smaller proportion of them (11%) were dopaminergic neurons belonging to the tuberoinfundibular cell group that inhibits prolactin secretion [228]. Indeed, KP inhibits prolactin secretion (Table 1.). However, prolactin is also a stress hormone that modifies the HPA response and has been implicated in postpartum depression [229], suggesting a role for KP in the etiology of this disease.

## 5. Coexpression of RFamides with Other Neurotransmitters

Since RFamide peptides act as neuromodulators in the CNS, they coexist at nerve endings with classical neurotransmitters and often with other neuromodulators (Table 2). The possibility of the release of classical neurotransmitters together with one or more neuromodulators ensures the plasticity of signaling [230]. Without this plasticity, the dynamic adaptation of the nervous system to constantly changing internal and external stimuli would not be possible [231,232]. In the reaction to stress, a classic example is the colocalization of CRH and AVP in PVN parvocellular cells with vesicular glutamate transporter 2 (VGLUT2), a marker of glutamate (the main excitatory neurotransmitter)-producing neurons [233,234]. In rats, chronic repeated restraint induced a remarkable plasticity in these cells, with a desensitization of the CRH response and an increase in the AVP response during HPA adaptation [18].

As shown above, RFamide peptides in the CNS contribute primarily to the integration of information related to the maintenance of homeostasis. Therefore, understanding the interaction between RFamide peptides and other neurotransmitters coexpressed in the same cell type is critical for unraveling the fine-tuning of homeostatic regulatory mechanisms. It also has potential benefits in elucidating the pathophysiology of various endocrine and psychiatric disorders, which may help in the development of new therapeutic strategies [235]. Unfortunately, there are only a limited number of studies available on the neurochemical nature of RFamide peptide-producing cells and the role of coexpression in these neurons.

**Table 2 cells-13-01097-t002:** Coexpression profile of RFamide peptide producing neurons.

RFamide Peptide	Area	Coexpression	Origin of Tissue/Cells	Method
**NPFF**	magnocellular PVN, SON	few cells, AVP	colchicine-treated male rats	single IHC, consecutive 10 µm-thick sections [236].
	rostral NTS	80% TH (adrenaline); 80% NPY; 20% cholecystokinin.	male mice	dual IHC; NPY-GFP transgenic mice/IHC [237].
	subpostrema	95% glutamate; 10% GABA/glycine.	mice	dual ISH, *VGLUT2*; dual ISH, *VGAT* [238].
	spinal cord laminae I-II	85% somatostatin; 38% *GRP*; 4.6% substance P.	male and female mice	dual IHC; dual ISH; dual ISH [239].
**RFRP**	hypothalamus	glutamate; galanin.	mice	single-cell RNA sequencing, *VGLUT2* [173]
	ARC	KP	OVX + estrogen rats	dual IHC [52]
	DMN	12% neurokinin B	male and female mice	dual ISH [240]
**PrRP**	NTS, ventrolateral medulla	all cells, TH (noradrenaline).	male rats male rats male and female rats	*PrRP* ISH/TH IHC [179]; dual IHC [241]; dual ISH [187].
	NTS & ventrolateral medulla	76% and 93% nesfatin-1/NUCB2	male rats	dual IHC [242]
	NTS, ventrolateral medulla	glutamate ~80% and ~16%, respectively.	male rats	*VGLUT2* ISH/TH IHC [243]
**QRFP**	PeN, medial preoptic area	77.9% glutamate; 7.2% GABA/glycine.	mCherry Q-hM3D transgenic mice	mCherry/*VGLUT2* or *VGAT* ISH [34]
	medial preoptic area	80% *BDNF*; 80% *PACAP*	mCherry Q-hM3D transgenic mice	mCherry/ISH [34]
	medial hypothalamus	glutamate orexin		single-cell RNA sequencing, *VGLUT2* [173]
**Kisspeptin**	ARC, KNDy neurons	all cells, dynorphin; 75% neurokinin B.	OVX + estrogen and ovary-intact ewes	dual IHC [244]
		96% dynorphin; 90% neurokinin B.	OVX +/− estrogen mice	dual ISH [245]
		75% neurokinin B.	post-mortem men	dual IHC [246]
		90% glutamate; 50% GABA.	KP-ß-galactosidase transgenic mice	ß-galactosidase IHC/*VGLUT2* ISH or *GAD-67* ISH [247]
	AVPV	33% dynorphin; 10% neurokinin B. 20% glutamate. 75% GABA.	OVX mice +/− estrogen male and female KP-beta-galactosidase transgenic mice	dual ISH [245] *GAD-67* ISH/ß-galactosidase IHC [247]

### 5.1. NPFF Peptides

NPFF was also detected via double IHC in some AVP neurons of the hypothalamic magnocellular cells in colchicine-treated male rats (Table 2) [236], supporting findings in rats and humans on the role of NPFF in the control of fluid homeostasis [248,249]. Nevertheless, AVP of both parvocellular and magnocellular origin can stimulate the HPA, with the potential for this effect to be moderated via NPFF [250].

In male mice, within the rostral NTS, 80% of NPFF neurons were double immunolabelled for tyrosine hydroxylase (TH), the rate-limiting enzyme of catecholamine synthesis, and FMRF amide peptide (detecting NPFF-like immunoreactivity). Using NPY–green fluorescent protein (GFP) transgenic mice, the authors reported that most of the NPFF-like neurons (80%) were GFP-positive [237]. Thus, in the rostral NTS, the majority of NPFF-like neurons coexpressed both NPY and TH, characteristic of C2 adrenergic cells [251] innervating the PVN (Table 2) [252]. It is noteworthy that the NPY system is implicated in stress-related neuropsychiatric disorders such as PTSD and depression. Furthermore, NPY may be a potential therapeutic target in protecting against the adverse effects of stress [253,254]. A minority (~20%) of NPY-GFP-negative NPFF-like neurons were immunopositive for cholecystokinin (Table 2) [237], a satiety-related peptide, which has also been associated with anxiety and depressive disorder [254].

Virtually all (95%) of the neurons expressing NPFF mRNA in the mouse subpostrema area, an autonomic regulatory center [255], were VGLUT2 -positive according to the double-RNAscope ISH technique [238]. A small percentage of NPFF neurons (~10%) expressed vesicular GABA transporter (VGAT), a marker of GABAergic and glycinergic neurons, suggesting that glutamate, GABA, and NPFF may be coexpressed in certain neurons (Table 2) [238,256,257]. 

The superficial dorsal horn (laminae I–II) is populated with excitatory interneurons classified according to their neuropeptide content. The area processes information related to pain, itching, and skin temperature [258,259]. NPFF neurons in the superficial dorsal horn were shown to be activated, for example, via noxious heat [258]. The majority (85%) of pro-NPFF-immunoreactive cells in the dorsal horn in mice (both sexes) contained somatostatin, which was detected via double IHC (Table 2). Interestingly, in the brain, somatostatin modulates anxiety, depression, stress, and fearful behavior [260], while in the spinal cord, it is associated with pain transmission [261]. However, pain and mood are bi-directionally related, increasing each other’s risk, suggesting a complex role for the somatostatin system in stress-related psychopathologies [261]. Furthermore, double-labeling fluorescence ISH demonstrated that 38% of the NPFF mRNA-producing neurons coexpressed gastrin-releasing peptide (GRP) mRNA (Table 2). GRP neurons are responsible for transmitting itching sensations in the spinal cord, and in the brain, they are involved in stress response [262,263]. Finally, a small percentage (4.6%) of NPFF mRNA-producing neurons coexpressed mRNA encoding substance P (Table 2). Further experiments have suggested that NPFF cells form a distinct population from cholecystokinin, neurotensin, or neurokinin B neurons [239].

### 5.2. RFRPs

RFRP neurons are excitatory neurons, according to single cell RNA sequencing data from mouse HTH (Table 2) [173]. Indirect data have confirmed the excitatory nature of RFRP cells. Gonad-intact female transgenic mice expressing tdTomato driven by vgat were immunonegative for RFRP-3, and immunomagnectically purified RFRP-3 cells from the HTH of male mice and female rats did not contain glutamic acid decarboxylase (GAD) mRNA, the rate-limiting enzyme of GABA synthesis [264].

Hypothalamic RFRP neurons coexpress galanin mRNA in mice (Table 2), which participates in the regulation of various homeostatic functions and is deeply involved in stress-related pathologies, including PTSD and depression [265,266].

RFRP-3 and KP immunoreactivities coexisted in ARC cells of OVX estrogen-primed rats (Table 2) [52], which is particularly interesting in the context of the generally opposing effect of these peptides on GnRH secretion [42,140,267]. In the DMN of adult mice of both sexes (both gonadectomized and intact animals), approximately 12% of neurons expressing RFRP coexpressed mRNA encoding neurokinin (NK) B protein [240]. The role of this protein, particularly in the DMN, remains poorly understood. However, the NK3 receptor (the cognate receptor of NKB) and the NK1 receptor have been shown to mediate the modulation of the dopaminergic, serotoninergic, and NA systems, which are affected by stress-related pathologies. Additionally, the NK3 receptor appears to influence the effects of cocaine, a psychostimulant drug [268].

### 5.3. PrRPs

Medullary PrRP cells belong to the A1 and A2 NA cell groups and therefore coexpress TH, which has been demonstrated via combined IHC and ISH [179], double ISH [187] and double IHC in rats (Table 2) [241]. The percentage of PrRP-positive cells out of the total number of TH-positive neurons was 82% in the A2 cell group and 98% in the A1 cell group, as shown via double IHC in rats [241]. PrRP enhances the effects of NA at several points, which has a great significance in response to stress [190,241].

Further characterization of rat PrRP neurons revealed that 76% of A2-PrRP cells and 93% of A1-PrRP cells also showed nesfatin-1/nucleobidin-2 (NUCB2) immunoreactivity (Table 2). Nesfatin-1/NUCB2 is also a stress molecule; it activates the HPA, and its expression is increased after restraint stress in the A1 and A2 cell groups [242]. The cooperative function of PrRP and nesfatin-1/NUCB2 in A1/A2 cell groups occurs during chronic stress, which may be important for maintaining the NA capacity of the cells [19].

PrRP neurons in the NTS are likely to be glutamatergic, as most A2 neurons (more than 80%) express VGLUT2 mRNA. Less is known about the major transmitter in A1 cells, a minority of which (approximately 16%) are VGLUT2-positive [243]. Unfortunately, no data are available on the chemical nature of PrRP neurons in the HTH (Table 2).

### 5.4. QRFPs

Regarding the classical neurotransmitter content, the majority (77.9%) of QRFP (mCherry Q-hM3D mice) neurons in the POA and periventricular region were excitatory neurons expressing VGLUT2 mRNA. However, a few cells (7.2%) were clearly inhibitory neurons producing VGAT mRNA. Therefore, like NPFF neurons, a small population (14.9%) of QRFP neurons probably express both glutamate and GABA [34]. The excitatory nature of the majority of QRFP neurons was confirmed via single-cell RNA sequencing (Table 2) [173].

The majority (80%) of QRFP (mCherry) neurons in the POA coexpressed mRNA encoding brain-derived neurotrophic factor (BDNF) and pituitary adenylate cyclase-activating polypeptide (PACAP) (Table 2) [34]. These cells are a special population of warm-sensitive neurons, and their optogenetic activation induces hypothermia [34,269]. In addition, the role of PACAP and BDNF in stress has been recognized. The molecule BDNF has been intensively studied as an antidepressant drug, and the BDNF gene is known as a common genetic locus of risk for mental illness [270,271]. Both PACAP and BDNF may therefore cooperate with QRFP in the reaction to stress. 

In addition, single-cell RNA sequencing has revealed that QRFP neurons in the medial HTH coexpress orexin (Table 2) [173]. Orexins are known to play a fundamental role in promoting arousal and wakefulness, a critical component of stress reaction, and dysfunction of the orexin system has been observed in PTSD, depression, and anxiety disorders [272].

### 5.5. Kisspeptins

KP-producing neurons in the ARC coexpress neurokinin B and dynorphin and are referred to as KNDy neurons (Table 2) [244,245]. KNDy neurons play a basic role in mediating the estrogen negative feedback and generate GnRH pulses. Whereas KP stimulates GnRH secretion, neurokinin B and dynorphin stimulate and inhibit the synchronized discharge of KNDy neurones in an autocrine/paracrine manner, respectively [245,273]. The KNDy hypothesis has become widely accepted as these neurons have been found in individuals of many mammalian species, regardless of sex [274]. However, human data have challenged this hypothesis. In post-mortem sections from young men, only about one third of neurokinin B neurons and 75% of KP neurons were double labelled with IHC [246]. In addition, a small percentage of KP neurons in the AVPV region also contain dynorphin or neurokinin B (Table 2) [245].

KP neurons form a heterogeneous population in terms of small neurotransmitter content. In transgenic mice (female and male) expressing GFP and β-galactosidase driven by the kisspeptin promoter, dual-label IHC/ISH showed that in the ARC, 90% of KP neurons (β-galactosidase-IR) coexpressed VGLUT2 mRNA, whereas 50% coexpressed GAD-67 mRNA, a marker of GABA neurons. In contrast, only 20% and 75% of KP neurons in the AVPV region were glutamatergic or GABAergic, respectively (Table 2) [247]. An interesting question is how these data relate to the fact that unlike in the ARC, KP neurons in the AVPV region of females are involved in the positive feedback of sex steroids [275].

## 6. Functional Role of RFamide Peptides Based upon Knockout (KO) Mice Models

Mice lacking NPFF exhibit normal body type, body composition, and locomotion and energy expenditure together with increased water intake and greater fuel-type flexibility under normal conditions. These mice have improved glucose tolerance, but their glucose homeostasis is more sensitive to diet-induced obesity than that of wild-type mice [238,276]. NPFF signaling also appears to be an important regulator of brown adipose tissue thermogenesis under challenging conditions such as a warm environment or high-fat diet [276]. In addition, both male and female npff KO mice show reduced repetitive behaviors and a decrease in anxiety-related behaviors [277].

Similarly, a deficiency in RFRP/GnIH also results in a decreased level of anxiety. Furthermore, KO mice showed decreased sensitivity to pain and performed intensive exercise in the dark phase compared with RFRP wild-type animals. Interestingly, the fertility data pertaining to the newly generated RFRP/GnIH KO animals were not presented in that study [160].

Due to the receptor promiscuity in the RFamide peptide family (Figure 2), the absence of NPFFR1 or NPFFR2 does not necessarily result in the same phenotype as the absence of RFRPs or NPFFs, which exhibit the highest affinity for these receptors, respectively. The lack of NPFFR1 in males kept on a high-fat diet caused a decrease in locomotor activity as well as impaired glucose tolerance and insulin sensitivity, whereas a high-fat diet or OVX in females led to obesity along with normal glucose homeostasis and reduced total energy expenditure [278]. Data are controversial regarding mice lacking NPFFR2. Npffr2 null mice maintained on a high-fat diet were reportedly obese [279], but these animals showed improved metabolic symptoms in mouse models of diabetes mellitus [280]. In harmony with the findings detected in ligand-deficient animals, NPFFR2-deficient animals were more resistant to stress-induced anxiety-like behavior than wild-type animals [281].

The leading symptoms in cases of PrRP or PrRPR deficiency are the development of late-onset obesity and metabolic syndrome in mice, with difference between the sexes [282,283,284,285], supporting the role of PrRP in homeostatic control. Receptor KO females are more affected than males due to their reduced energy expenditure [283]. Mutations in the prrpr gene have recently been linked to obesity in humans as well [286]. Studies in prrp KO mice have confirmed that PrRP also plays a role in pain modulation. The absence of PrRPR led to an increase in the basal nociceptive threshold, stress-induced analgesia, and the reinforcing effect of morphine compared with wild-type mice [74]. PrRPR deficiency also manifests in a phenotype that is less anxious [287].

Qrfp null mice are hypophagic and have impaired glucose homeostasis, but are lean, probably due to their elevated basal metabolic rate [288,289]. These animals also show increased anxiety-like behavior and disruption of circadian rhythm [288].

Interestingly, both qrfpr1 and qrfpr2 KO mice have a normal metabolic phenotype, and the central effects of QRFP on food intake and locomotion require the presence of both QRFPR orthologs [32].

The essential role of KPs in reproduction is evidenced through the fact that the absence of kp/kiss1r genes causes hypogonadotropic hypogonadism in both rodents and humans [140]. In addition, female mice lacking the kiss1r gene exhibited obesity, impaired glucose tolerance, reduced energy expenditure and food intake, and impaired thermoregulation, while males exhibited a less severe phenotype at 22°C [290,291]. However, reduced anxiety in receptor KO male mice was observed in the elevated plus maze test, which was further enhanced through the restoration of testosterone levels [292].

## 7. RFamide Peptides in Stress and Stress-Related Diseases

Although general knockout models cannot distinguish between the peripheral and central effects of the missing gene product, the overall findings, together with the above described localization, projection, and coexpression data, confirm a critical role of RFamide peptides in homeostatic regulation and highlight their potential role in stress-related pathologies. A summary of the central effects of RFamide peptides on the HPA/SAM and related disorders is presented below.

### 7.1. NPFF Peptides

Although members of the NPFF family of peptides were originally thought to act as anti-opioid neuromodulators in modifying pain perception [67], their role in the stress response is now being outlined. 

NPAF and NPSF activate the HPA. Both increased plasma ACTH and CORT levels in rats 10 min after ICV administration, which could be prevented via pretreatment with a CRH receptor antagonist, suggesting an effect through the CRH-producing parvocellular PVN [24,293]. The CRH neurons within the PVN are controlled by a tonic GABAergic inhibition from the peri-PVN zone and the BNST. These GABAergic neurons are regulated by stress-sensitive brain areas deeply involved in the pathomechanism of depression, anxiety, and PTSD, such as the medial prefrontal cortex, AMY, and HC [294,295]. It is also known that hyperactivity of the HPA, which is always observed in stress-related mental illnesses and is induced by chronic stress [296], impairs GABAergic inhibition of the PVN [297]. It is, therefore, particularly interesting that NPFF has been shown to disinhibit the GABAergic input to parvocellular PVN neurons [298].

NPFF is also a central activator of the SAM. Both haemorrhage and hypertension activate NPFF neurons in the NTS. When injected ICV, intrathecally (IT), or directly into the NTS, which collects cardiovascular inputs, NPFF increases blood pressure and heart rate [299]. Central NPFF treatment induces cell activation primarily in parvocellular pre-autonomous oxytocin-containing neurons in the PVN [300], which have a well-established function in sympathetic regulation of the heart, blood vessels, and kidneys [301]. Human data also support the involvement of NPFF in the cardiovascular regulation. NPFF innervation of the HTH was found to be dramatically reduced in hypertensive patients compared with controls [248].

In a series of behavioral tests performed using rats and mice, NPAF dose-dependently increased anxiety, consistent with its effect on HPA activity [24,302]. The anxiogenic effect of NPAF was attenuated through pretreatment with the specific CRH receptor 1 antagonist antalarmin [24]. In contrast, the antidepressant-like effect of NPAF in mice has also been demonstrated in modified forced swim test experiments, as ICV administration of NPAF reduced immobility time and increased climbing and swimming times. The serotonin 1 and 2 receptor antagonist methysergide completely reversed this effect of NPAF, suggesting a serotonergic neurotransmission-mediated mode of action [302].

While ICV administration of NPFF to rats did not affect anxiety behavior in the elevated plus maze test [81], NPFF enhanced anxiety-like behavior during withdrawal from chronic ethanol administration via an interaction with the opioid system [303]. On the other hand, chronic mild stress upregulated NPFF mRNA expression in the medial prefrontal cortex, HC, AMY, and HTH [304].

A large body of data supports a role for NPFFRs in affective disorders. However, the broad affinity of these receptors for different RFamide peptides often makes it difficult to identify the neural pathways and the transmitters involved. Amphetamine withdrawal evoked anxiety-like behavior in rats, which was reversed with the central addition of NPFFR1/2 antagonist RF9. In addition, the effect of RF9 was attenuated via NPFF [305]. Silencing of NPFFR2 expression via shRNA in the PVN inhibited the development of chronic mild stress-induced depression-like behavior in mice [304]. In contrast, different NPFFR2 agonists applied ICV or intraperitoneally were able to activate the HPA in rats and mice, which was reflected in increased levels of ACTH and CORT levels [304,306]. In npffr2 transgenic mice, neuron-specific enolase promoter-driven overexpression of NPFFR2 resulted in effects similar to chronic activation of NPFFR2. These effects included reduced ability to cope with stress, increased anxiety-like behavior and anhedonia, hyperactivation of the HPA, and reduced expression of hippocampal glucocorticoid receptors [304]. The overall data suggest a potential beneficial effect of NPFFR2 antagonism in affective disorders.

### 7.2. RFRPs

The acute and chronic stimulatory effects of RFRPs on the HPA have been confirmed in male rodents [93,114]. In mice, both acute and chronic central administration of RFRP-3 resulted in elevated cortisol levels, which were prevented using GJ14, a selective NPFFR1 antagonist [114]. The mechanism of action is either direct or indirect. In adult OVX ewes, RFRP-IR fibers formed close contact with approximately 30% of CRH-IR cells in the PVN, suggesting a direct action on these cells [163]. In contrast, ex vivo electrophysiological experiments in rats indicated that NPVF (the human RFRP-3) disinhibited neuronal activity in the parvocellular PVN in a manner similar to NPFF [298]. Either way, ICV administered RFRP-3 activated the majority of CRH neurons (i.e., increased cFos-positivity, a marker for neuronal activation) in CRH-GFP male transgenic mice via NPFFR1s [114].

In addition to the PVN, centrally injected RFRP1 and RFRP3 were found to activate other hypothalamic nuclei involved in the organization of responses to stress: the SON, PeN, and ARC [158]. Moreover, in both male and female rodents, RFRP neurons in the DMN were activated (i.e., cFos positivity) via various physical and psychosocial stressors, such as foot shock or acute restraint [93,307,308]. Since there are no reuptake mechanisms for neuropeptides, cell activation and the subsequent neuropeptide release must be followed by de novo synthesis of the neuropeptides in the cell body [17]. Accordingly, RFRP mRNA levels were increased 3 h after acute restraint in the DMN, indicating previous RFRP release [307,308]. Glucocorticoids appear to exert positive feedback on RFRP neurons, 53% of which express glucocorticoid receptors, as adrenalectomy abolished the increase in RFRP mRNA in response to acute stress. It is notable that a small percentage of RFRP cells also express CRF receptor 1 [307].

The contribution of RFRPs to the development of anxiety-like behavior has also been investigated. Anxiogenic effects were observed with acute central administration of RFRP-1 or RFRP-3 in male rats [93] and chronic infusion of RFRP-3 ICV in male mice [114]. Importantly, chronic infusion of the NPFFR1 antagonist GJ14 induced anxiolytic effects, and coinfusion of RFRP-3 and GJ14 reversed the effects of RFRP-3 on anxiety-like behavior [114]. Furthermore, RFRPs stimulated the NA-dopaminergic LC [158] known to be involved in arousal, anxiety, depression, and cognitive processes. LC is also fundamental in driving the comorbidity of pain and stress-related mental disorders [309].

Although there is no direct evidence in the literature linking RFRPs to the pathophysiology of PTSD, it is notable that RFRPs administered ICV activated the nucleus incertus [158]. This nucleus innervates central hubs orchestrating various aspects of the stress response [310]. In particular, it is critical in the control of formation of contextual fear memories; therefore, it is probably a basic structure in the pathomechanism of PTSD that is characterized by emotional hypermnesia [311]. 

Since RFRPs generally act as GnRH inhibitors [312], the fact that the RFRP gene is also upregulated in chronic stress situations is of particular importance [307,313]. Reproduction is essential for the survival of species, and GnRH secretion is severely impaired in chronic stress [308]. A series of experimental data derived from adult male and female rats suggests that RFRPs may provide the molecular basis of this phenomenon via inhibiting the HPG [307,313]. Early-life stress can also have long-term negative consequences on reproduction through a similar mechanism. Neonatal glucocorticoid treatment of female mice resulted in delayed puberty and reduced GnRH mRNA levels. In parallel, RFRP mRNA expression in the DMN and NPFFR1 mRNA expression in the POA were upregulated. Overall, the data indicated a stress-induced adaptive change in the RFRP-GnRH pathway, which had a detrimental effect on the development of the reproductive system [314].

### 7.3. PrRP

The regulation of stress responses has emerged as a major function of PrRP in mammals. As described above, NA and PrRP are coexpressed in the A1 cell group and in the A2-NTS area in the medulla oblongata [181]. The A1 and A2 neurons are the main sources of NA afferents to the PVN [315]. The integrity of this ascending pathway is indispensable for the stimulation of the HPA via homeostatic stressors and peripheral inflammation [316]. Stress-related information from the body is carried through the vagus nerve to the NTS, where vagal afferents directly innervate the A2 NA cells [317]. Therefore, the PrRP-NA coexpressing neurons in the NTS are in a gating position between the vagus-mediated visceral signals and the brain. However, the medullary PrRP-NA neurons and the non-catecholaminergic PrRP cells in the DMN contribute almost equally to the innervation of different hypothalamic regions, including the PVN [19]. This suggests that PrRP neurons in the DMN play an important role in homeostatic regulation, including the control of stress response.

In rats, PrRP administers ICV stimulated the release of ACTH [241] and CORT [318], acting directly on CRH neurons in the PVN [319,320] and/or acting on BNST neurons that in turn disinhibited CRH neurons [130,194,197]. Alterations in the function of the HPA were also revealed in PrRP-deficient and PrRPR-deficient mice, confirming the data for exogenously administered PrRP [74,321,322]. Importantly, PrRP enhanced the ACTH-stimulating effect of NA, thus influencing the efficiency of NA signaling [241,323]. Cooperation of coexpressed neurotransmitters has a relevance in adaptation to chronic stress [18], indicating a main role of PrRP in chronic stress reaction. In fact, the PrRP-TH ratio shifts in favor of PrRP in the A1/A2 cell groups during chronic restraint and chronic osmotic challenge, supporting this idea [19,187]. Furthermore, medullary PrRP neurons express estrogen receptor alpha, and PrRP expression in the medulla changes differently in response to chronic restraint stress in male and female rats [185,187]. The reaction to stress and the prevalence of stress-related mental illnesses in humans also show a sex dependence [2]. PrRP may therefore be a key molecule in mediating the sex-dependent effects of stress in the brain.

In addition to the HPA, PrRP also activates the sympathetic nervous system [324]. When injected into the caudal ventrolateral medulla, PrRP increased the mean arterial pressure, heart rate, and renal sympathetic nerve activity [325]. Furthermore, PrRP cells in the NTS showed reduced PrRP immunoreactivity before and during the development of hypertension in spontaneously hypertensive rats [326], which may have been a sign of a prolonged release. 

Medullary PrRP neurons react to acute and chronic stress by activating cells and upregulating PrRP mRNA [327]. Both physiological stressors [131,190,241,328] and mixed physiological–psychogenic and emotional stressors recruit medullary PrRP neurons [19,187,329,330]. Furthermore, PrRP neurons in the DMN also react to different kind of stressors by activating cells and upregulating PrRP mRNA [133,187,241,321].

The fundamental role of PrRP in the control of stress reaction suggests its involvement in the development of stress-related mental disorders. Indeed, in a PTSD model, reexposure to a conditioned fear stimulus failed to increase plasma ACTH in PrRP-deficient mice [321]. Intranasal application of PrRP increased anxiety and decreased sociality in male rats [331], while PrRPR-deficient mice showed reduced anxiety-like behavior [287]. Recently, the role of PrRP in the pathomechanism of depression has also been emphasized. Using different animal models of depression, forced swim testing, learned helplessness, and peripheral inflammation, researchers have provided evidence that chronic stress leads to overload of the PrRP system, resulting in impaired coping with stress. Depression-vulnerable animals show signs of insufficient PrRP signaling in the dorsolateral HTH, characterized by reduced density of PrRP-IR axons, downregulation of PrRPR and NPFFR2, and dysregulation of MCH expression in the local population [190]. MCH neurons serve as a key hub for regulating affective disorders [166]; therefore, inadequate control of MCH neurons via PrRP can be assumed as a possible pathomechanism. This hypothesis has been supported by both ex vivo electrophysiology and in vivo animal experiments, which confirmed that PrRP inhibited MCH neurons. Furthermore, in patch clamp experiments, PrRP enhanced the inhibitory effect of NA on MCH cells. The expression of PrRPR and NPFFR2 was also reduced in the dorsolateral HTH of suicidal subjects, highlighting that the fine-tuning of MCH activity via PrRP may be relevant in the patomechanism of human depression as well [190].

### 7.4. QRFPs

The implication of QRFP peptides in stress behavior was proposed due to rich QRFPR1 and QRFPR2 mRNAs expression in rodent brain regions involved in stress and anxiety [31,96,107,201]. However, the effects of QRFP on the HPA function have not been confirmed in vivo, and central injection of QRFP failed to elicit cFos expression in the PVN of mice [32]. However, in vitro, a QRFP-induced increase in crh promoter activity and CRH expression was observed in hypothalamic 4B cells with parvocellular PVN neuronal characteristics [203]. 

QRFP may be also involved in the stress response via activating the SAM. Centrally administered QRFP caused rapid and massive increase in blood pressure and heart rate in mice [46]. In the same study, QRFP induced intensive grooming in mice, a marker of stress and anxiety, although anxiety-like behavior was not observed in the elevated plus maze test. Similarly, negative behavioral test data were obtained when QRFP was injected into the medial HTH of male rats [86]. However, the effect of QRFP on anxiety is controversial. The ICV administered synthetic QRFPR agonist P550 peptide exerted an anxiolytic effect in mice, which was completely abolished via phenoxybenzamine (a non-selective α-adrenergic receptor antagonist) and bicuculline (a competitive antagonist of GABA_A_ receptors) [332]. The anxiety-like behavior of QRFP-deficient mice confirmed these data [288].

On the other hand, QRFP increased cFos expression in orexin neurons in the LHA in mice [32], which also happens in response to acute stress [272]. Interactions between the QRFP and the orexin systems have been demonstrated at multiple levels [32,204,333], providing a basis for the possible contribution of QRFP to the organization of certain aspects of the stress response through the orexin system.

### 7.5. Kisspeptins

Given the critical role of KPs in fertility and the deleterious effects of stress on reproduction, the reciprocal interaction of the KP system with stress brain centers is essential for species maintenance. 

KP-13 and KP-8 stimulated the HPA when administered ICV in male rats. This was reflected in an increase in the plasma CORT levels, which could be blocked by both CRH receptor antagonist and AVP receptor-1 antagonist [35,95]. However, in vitro, in hypothalamic PVN-derived cell lines, KP increased the expression of AVP mRNA but decreased the expression of CRH mRNA, albeit only at high doses [334]. The ability of KP to induce AVP release in vivo has been confirmed in several studies [35,95,335]. Although AVP released from the magnocellular PVN and SON neurons regulates blood volume, parvocellular AVP coexpressed with CRH stimulates the HPA [2]. KP increased AVP secretion in response to volume tension, without activating the SAM [335]; therefore, it is possible that KPs induced both parvocellular and magnocellular AVP release. Furthermore, KPs influenced CRH and AVP signaling in the AMY and HC [35], which are key structures in stress-related memory formation and are deeply involved in stress-related neuropathologies [295]. 

In line with the stimulation of the HPA, adult rats injected with KP-8 or KP-13 ICV spent more time in the closed arms of the elevated plus maze [35,95], whereas kiss1r KO mice preferred the open arms compared with controls, suggesting an anxiogenic effect of KP [292]. However, after selective stimulation of KP neurons in the posterodorsal medial AMY of male mice via DREADDs (designer receptors exclusively activated by designer drugs), mice exhibited anxiolytic-like behavior [336]. Higher doses of KP-13 given ICV also had a beneficial effect on the ability to cope with stress in the forced swim test [337], which could be blocked by the nonselective α-adrenergic receptor antagonist phenoxybenzamine, the α2-adrenergic receptor antagonist yohimbine, and the nonselective serotonin receptor 2 antagonist cyproheptadine [337]. Therefore, the data indicate a dose-, site- and situation-dependent effect of KPs on stress responses and suggest a possible interaction between KPs and adrenergic/serotoninergic systems. 

Furthermore, Comninos et al. al [338] used functional magnetic resonance imaging to evaluate the effects of intravenously administered KP on brain function in heterosexual young men. KP administration enhanced limbic responses to sexual and bonding stimuli, improved positive mood, and attenuated negative mood [338]. KP administration also modulated resting brain connectivity to enhance emotional processing [339]. Experiments on mice demonstrated that peripherally injected radiolabeled KP was able to gain access to cortical, limbic, and other brain structures. Peripheral KP administration did not alter blood concentrations of testosterone, oxytocin, or CORT [338], and the latter was shown also in rats [334]. Overall, it appears that centrally acting KP plays a modulatory role in stress response and recruits extrahypothalamic, mainly limbic areas.

Although the exact mechanisms of the effect of KP under stress remain to be clarified, the negative impact of stress on the KP/Kiss1R system and the reproductive axis is supported through a substantial body of evidence. For example, various types of stressors, including lipopolysaccharide-induced inflammation, acute restraint, or insulin-induced hypoglycemia, disrupted LH pulsatility, decreased KP expression or KP neuron activity, and altered Kiss1R mRNA expression in the HTH of female rodents [308,340,341]. Unpredictable chronic stress also reduced KP immunoreactivity in the HTH of male mice [342]. Optogenetic experiments revealed that the CRH neurons in the PVN inhibit KNdy neurons in the ARC via GABA interneurons [343]. Indeed, KP and Kiss1R expression was reduced via centrally applied CRH in both the AVPV and ARC [341]. The administration of exogenous CORT has been demonstrated to mimic the negative effects of stress on the KP/Kiss1R system, providing evidence that these effects are a consequence of HPA activation [341,344].

## 8. Summary and Future Perspectives

A great body of evidence supports the emerging role of RFamide peptides in stress and stress-related psychopathologies. The cognate receptors of all RFamide peptides are abundantly expressed in the HTH, the main autonomic and endocrine integration center of the brain. Limbic structures (BNST, septum, hippocampal formation) and brainstem autonomic centers (LC, raphe nuclei, NTS, DMX, AP) involved in adaptation to stress also express multiple types of RFamide peptide receptors (Figure 3). With regard to receptors, it is essential to acknowledge that, in addition to their cognate receptors, all RFamide peptides exhibit some degree of affinity for NPFFRs (Figure 2) [63]. This may explain some discrepancies between the distribution of the immunoreactive fibers and the cognate receptors of RFamide peptides and may rise a number of interesting questions about RFaminde signaling.

Nevertheless, regarding HPA, a common feature of RFamides is that they all can stimulate it, although in the case of QRFP, this is only supported with in vitro data (Figure 4). In addition, sympathetic effects of NPFF, PrRP, and QRFP have also been demonstrated (Figure 4). Furthermore, most RFamide peptides affect anxiety-like behavior and depression-like behavior, and PrRP appears also to be involved in the pathogenesis of PTSD. Interestingly, anxiety is increased while depressive-like behavior is reduced via certain RFamide peptides (Figure 4). This indicates that anxiety and depression develop differently, and the effects of RFamides may be dose-, site-, and situation-dependent. Although anxiety disorders and depression share a high degree of comorbidity, similar findings were described in elastase-2 KO and BDNF transgenic mice [345,346], and increased anxiety during the initial phase of antidepressant treatment is a common side effect. 

Homeostatic threats cause stress and stress disrupts homeostasis. Thus, another important point is that RFamide peptides are involved not only in the stress response but also in the regulation of basic homeostatic parameters. Certain RFamide peptides may have their own niche in the system. For example, NPFF peptides may represent a primary link between pain and stress [63,69,113], while KPs and RFRPs connect stress to reproduction/fertility [217,347]. Furthermore, sleep disturbances are among the leading symptoms associated with stress-related psychiatric diseases [348]; therefore, the interaction between stress and circadian rhythm also holds a particular interest. The lack of QRFP results in a disruption of the circadian rhythm [288], indicating that QRFP may be a key molecule that mediates this interaction. The close relationship between the QRFP and the orexin systems support this putative role of QRFP [173,204,333]. However, the PrRP-MCH connection [190] suggests a further link between RFamide peptide and sleep–wake regulation, since MCH neurons are essential in promoting and maintaining sleep [349]. In addition, RFamide peptides are all involved in the regulation of energy balance [350], which is profoundly affected by stress. There is a high comorbidity of depression with chronic pain [309] and eating disorders [351]. Moreover, chronic stress has also been implicated in the etiology of certain reproductive diseases (e.g., endometriosis, polycystic ovarian syndrome) that cause infertility [352,353]. RFamide peptides could potentially be targets for therapeutic intervention for these specific problems.

Although this review has concentrated on the central effects of RFamide peptides, it should be noted that circulating RFamide peptides may also contribute to the stress response via the HPA at the periphery. It has been shown that mRNAs encoding PrRPR, NPFFRs, KISS1R, RFRP, and QRFP are expressed in the human pituitary gland [101,109,125,196,205]. PrRP, NPFF, and their receptors have also been detected in the human adrenal gland. Additionally, QRFP immunopositivity is particularly dense in the zona fasciculata, where CORT is produced [101,128,196,205,354].

Different RFamide peptide analogues already exist and their effects are being continuously tested in animal models [355,356,357]. However, further research is needed to elucidate the exact mechanisms of their action. The characterization of the chemical profiles of the different RFamide cell populations is an essential step in this direction and provides a unique opportunity to design effective, specific, and side-effect-free therapeutic peptide cocktails in the future.

## Figures and Tables

**Figure 1 cells-13-01097-f001:**
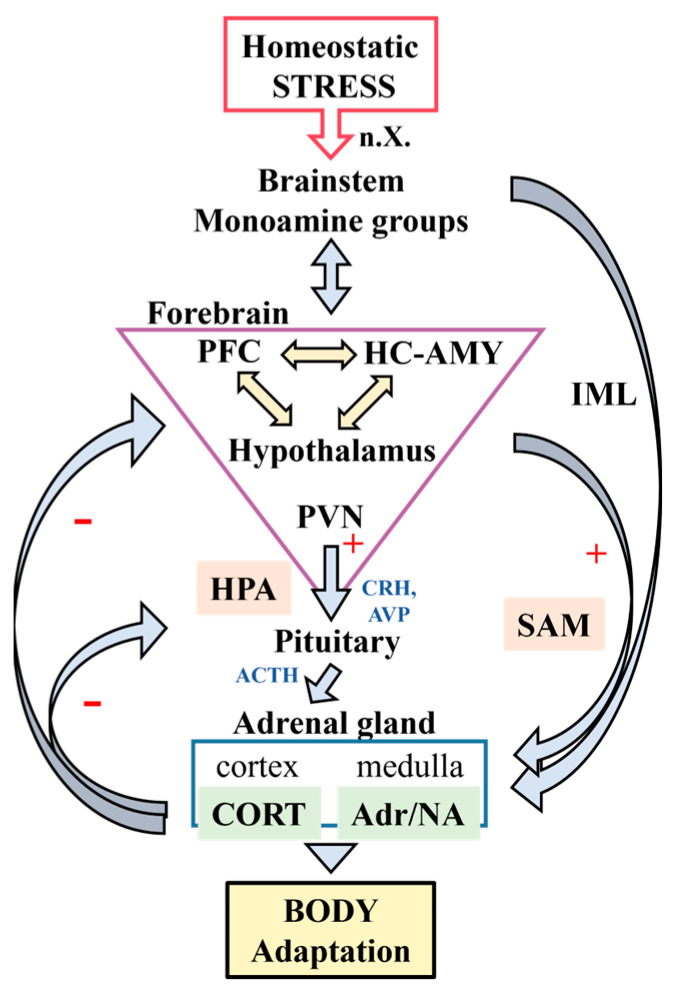
Schematic representation of the basic organization of the stress response to homeostatic stressors. ACTH: adrenocorticotropic hormone; Adr: adrenaline; AMY: amygdala; AVP: arginine vasopressin; CORT: glucocorticoids—in humans, cortisol, in rodents, corticosterone; CRH: corticotropin-releasing hormone; HC: hippocampus; HPA: hypothalamic–pituitary–adrenal axis; IML: intermediolateral cell column of the spinal cord; NA: noradrenaline; n.X.: nervus vagus; PFC: prefrontal cortex; PVN: hypothalamic paraventricular nucleus, the center of the HPA; SAM: sympathoadrenomedullary system.

**Figure 2 cells-13-01097-f002:**
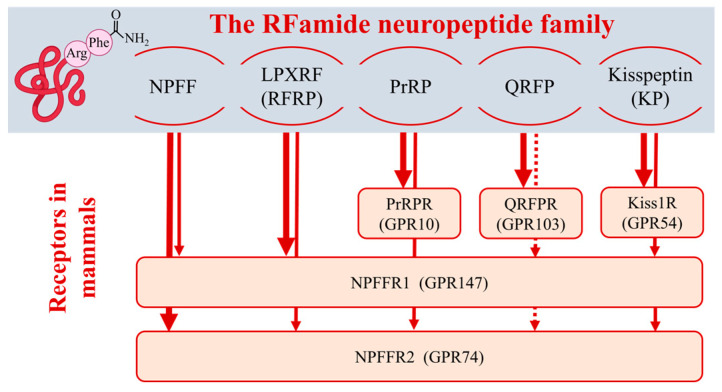
The RFamide peptide family and their receptors. Thick arrows: high affinity at the receptor; thin arrows: lower affinity at the receptor with biological activity; dashed arrow: biological activity is controversial. For references, see the text.

**Figure 3 cells-13-01097-f003:**
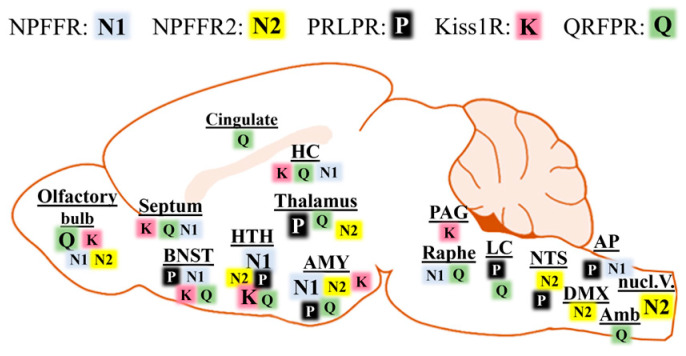
Distribution of RFamide peptide receptors in brain areas associated with stress reaction in rodents. The figure shows areas of high and moderate receptor expression based on ISH data. For references, see the text. Amb: ambiguous nucleus; AMY: amygdala; AP: area postrema; BNST: bed nucleus of stria terminalis; cingulate: cingulate cortex; DMX: dorsal motor nucleus of the vagus nerve; HC: hippocampus; HTH: hypothalamus; LC: locus ceruleus; NTS: nucleus of the solitary tract; nucl.V.: spinal trigeminal nucleus, PAG: periaqueductal grey matter; raphe: raphe nuclei.

**Figure 4 cells-13-01097-f004:**
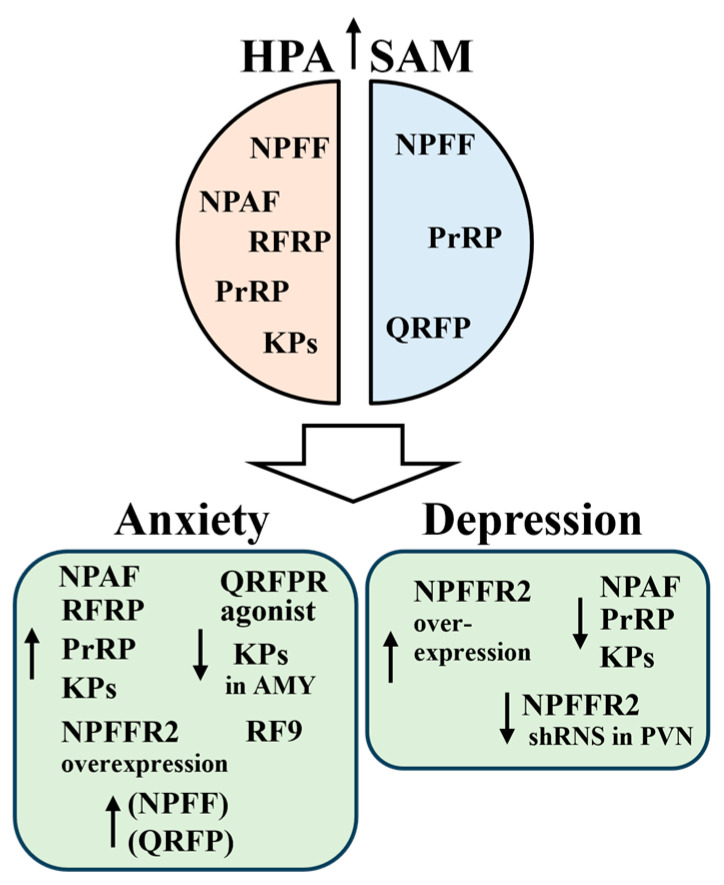
Summary of the effects of RFamide peptides on HPA and SAM activity and various stress-related disorders. Data are based on administration of the drugs ICV, unless otherwise indicated. Parentheses are used to indicate that the data were dependent on the behavioral test used. RF9: NPFFR1/2 antagonist. For references, see the text.

**Table 1 cells-13-01097-t001:** The effects of RFamides on energy balance, reproduction, pain perception, reward, learning, and activity. Data are based on central administration of RFamides and in vivo stimulation of RFamide-producing neurons. ARC: arcuate nucleus; AMY: amygdala; AVPV: rostral periventricular region of the third ventricle; CPP: conditioned place preference; DMN: dorsomedial hypothalamic nucleus; ICV: intracerebroventricular; IT: intrathecally; KP: kisspeptin; LC: locus coeruleus; LH: luteinizing hormone; NTS: nucleus of the solitary tract; PAG: periaqueductal gray; PBN parabrachial nucleus; POA: preoptic area; PRL: prolactin; VTA: ventral tegmental area.

Effect	NPFF/NPAF	RFP1/RFP3	PrRP	QRFP	Kisspeptins
**Energy** **expediture**	**Hypothermia:** NPFF, ICV [22,23]; NPAF, ICV [24].	**Hypothermia:** RFP3, ICV [25,26,27]	**Hyperthermia:** ICV [28], brief hypothermia then long-lasting hyperthermia [29]; Fourth ventricle, NTS, hyperthermia [30].	**No effect:** ICV [31,32]; **Hypothermia, reduced thermogenesis:** ICV, chronic treatment [33]; **Hypothermia and hibernation-like state:** chemogenetical activation [34].	**Hyperthermia:** KP-13, ICV [35].
**Food intake**	**Anorexigenic:** NPFF, ICV [36,37]; NPAF, ICV [38]. **Orexigenic:** NPFF, lateral PBN, high dose [39].	**Anorexigenic:** RFP1/3, central AMY [40,41]; **Orexigenic:** RFP3, ICV in the light phase [42], chronic infusion [27].	**Anorexigenic:** ICV [28,29]; DMN [43]; NTS [30].	**Orexigenic:** ICV [32,44,45,46,47], high fat intake [48]; medial hypothalamic area [49].	**Anorexigenic:** KP-10, ICV [50].
**Reproduction**		**Inhibits GnRH cells:** RFP-3, ICV [51]; **Inhibits KP cells:** RFP-3, ICV [52,53]; **Inhibits LH secretion:** RFP-1/3, ICV [52,54,55]; **Stimulates LH secretion in males:** RFRP-3, ICV [54]; **Stimulates PRL release:** RFRP-1, ICV [56].			**Stimulates LH secretion:** KP-54/10, ICV [57]; KP-10, ARC, POA [58]; medial AMY [59]. **Stimulates FSH secretion:** KP-54, ICV [57]; **Stimulates PRL secretion:** KP-10, ICV [60]; **Stimulates sexual behavior:** KP-10, ICV [61]; photostimulation, AVPV [62]; KP-10, medial AMY [59].
**Pain perception**	**No effect:** NPFF, ICV, on basal nociceptive threshold [63]. **Antinociceptive:** NPFF, PAG, antiallodynia [64]; NPFF, NPAF, NPSF, IT, analgesic effect, enhanced morphine analgesia [65,66]. **Nociceptive** NPFF, ICV, VTA, PAG, hyperalgesia, reduced morphine and stress analgesia [23,64,67,68,69,70,71]; NPSF, ICV, reversed morphine analgesia [70].	**Antinociceptive:** RFP1, IT, antiallodynia, antinociception [72]; RFP3, ICV, enhanced morphine analgesia [25]. **Nociceptive:** RFP3, ICV, reduced warm-water-swim stress-induced analgesia [69]; reduced basal nociceptive threshold [63].	**Antinociceptive:** PAG, antiallodynia; NTS, antinociception [73]. **Nociceptive:** ICV, reduced basal nociceptive threshold and morphine analgesia; [63,74] caudal ventrolateral medulla, hyperalgesia [73]. **No effect:** IT [73].	**Antinociceptive:** ICV, IT, antiallodynia [75] and analgesia [76,77]; LC, PAG, analgesia [78]. **Nociceptive:** ICV, reduced basal nociceptive threshold [63]	**Nociceptive:** KP-10/13, ICV, reduced basal nociceptive threshold and morphine analgesia [63,79]; KP, IT, hyperalgesia [80]
**Reward**	**Negative effect:** NPFF, ICV, anti-opioid effect in CPP test [81,82].	**Positive reinforcement:** RFP1, central AMY [83].	**No effect:** 4th ventricle, food reward [30].	**No effect** ICV, food reward [32]	
**Learning/** **Memory**	**Improved learning:** NPAF, ICV, reversed memory impairment [84].	**Improves learning:** RFP1, central AMY [85].		**Improves memory** Medial hypothalamic injection [86]	**Improves memory and learning:** reversed memory impairment KP-13, ICV [87,88,89]; KP-13, hippocampus [88];
**Locomotion**	**Hypoactivity:** NPFF, ICV, anti-opioid effect [82,90]; VTA, anti-opioid, anti-novelty effects [91,92]. **Hyperactivity:** NPAF, ICV [24].	**Hypoactivity:** RFP3, ICV, decreased total locomotion [93]. **No effect:** RFP3, chronic infusion ICV [27].	**No effect:** ICV, repeated injection, measured on day 3 [94].	**Hyperactivity** ICV [32,46,47] **No effect** ICV [31] **Hypoactivity** Chemogenetical activation, long-lasting effect [34]	**Hypoactivity:** KP-8, ICV [95]. **Hyperactivity:** KP-13, ICV [35].

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
