# Peer review of "Brain RFamide Neuropeptides in Stress-Related Psychopathologies"

_cells, 2024, doi:10.3390/cells13131097_

Round 1

Reviewer 1 Report

Comments and Suggestions for Authors

Kovacs et al. reported in this submitted article a thorough review of RFamide system in stress-related physiology and pathology. Authors provided a detailed description in regard general features of 5 classes of RFamide neuropeptide as well as their corresponding receptors and focused on their neuroanatomy and co-expression with designated neurotransmitters. Also, authors brought in a functional role of these neuropeptides based on knockout animal models and experimental results of stress-related responses. Overall, this is a well-organized article and authors apparently provide knowledgeable information that updates the recent findings of RFamide, focusing on the HPA axis and SAM-mediated stress. I have no problem to accept this manuscript for publication, but have some comments as listed below:

1.      Under the Introduction, I would suggest authors to add a subheading of “stress and stress-related neuro-circuitries” to link with following RFamide themes.

2.      The co-expression with neurotransmitter is interesting, but considering that neuropeptide might mainly act through their receptor to affect the projected neurons and participate body stress response. The review of co-expression of receptor with targeted neurons would be more functionally significant.

3.      There are reports that indicate kisspeptin would regulate prolactin release mediated through hypothalamic DA neurons. I wonder authors could comment on the kisspeptin with involved catecholamines?

4.      NPFF could functionally modulate the DA-mediated brain functions (as indicated the locomotion and mood), I wonder if NPFF would also synapse with DA neurons, rather than co-expressed with C2 neurons?

5.      This review seems to pretty much focus on the central effect of RFamide, however peripheral stress response could also be affected by circulation RFamides. I would appreciate if authors could make brief comments on their peripheral role.

Author Response

We thank the reviewer for her/his work and valuable thoughts. The following is a point-by-point response to the comments raised.

Under the Introduction, I would suggest authors to add a subheading of “stress and stress-related neuro-circuitries” to link with following RFamide themes.

It is an excellent idea. Subheadings have been added to the Introduction (see the modifications at lines 33, 97,116).

  1. The co-expression with neurotransmitter is interesting, but considering that neuropeptide might mainly act through their receptor to affect the projected neurons and participate body stress response. The review of co-expression of receptor with targeted neurons would be more functionally significant.

The two types of co-expression analysis yield disparate yet equally informative data, both of which are of functional interest. The coexpression of cells producing RFamide peptides is summarized in one chapter, while the chemical nature of neurons coexpressing RFamide receptors was described under the heading "Receptors" in the original manuscript. Due to the limited availability of data, this topic is not addressed in a separate chapter. Nevertheless, to emphasize the significance of this topic, it is addressed in a separate subheadings in the revised version. Furthermore, we conducted additional literature searches and incorporated new data into the text (lines 383-401, 449-459, 510-512,574-585).

  1. There are reports that indicate kisspeptin would regulate prolactin release mediated through hypothalamic DA neurons. I wonder authors could comment on the kisspeptin with involved catecholamines?

Indeed, this finding is included in Table 1. There is a morphological basis for this effect of kisspeptin, as about 15% of tuberoinfundibular dopamine (TIDA) neurons express Kiss1R, albeit at low levels [1]. This effect of kisspeptin may also be relevant to stress-related disorders, as prolactin is involved in modifying the HPA response and has been implicated in postpartum depression [2]. We have added this information to the text (lines 579-585).

  1. Higo, S.; Iijima, N.; Ozawa, H. Characterisation of Kiss1r (Gpr54)-Expressing Neurones in the Arcuate Nucleus of the Female Rat Hypothalamus. J Neuroendocrinol 2017, 29, doi:10.1111/jne.12452.
  2. Faron-Górecka, A.; Latocha, K.; Pabian, P.; Kolasa, M.; Sobczyk-Krupiarz, I.; Dziedzicka-Wasylewska, M. The Involvement of Prolactin in Stress-Related Disorders. Int J Environ Res Public Health 2023, 20, doi:10.3390/ijerph20043257.
  3. NPFF could functionally modulate the DA-mediated brain functions (as indicated the locomotion and mood), I wonder if NPFF would also synapse with DA neurons, rather than co-expressed with C2 neurons?

A search of the literature has not revealed data on synaptic connections between NPFF-positive fibers and dopaminergic cells. However, neuropeptides often act non-synaptically, by volume transmission. Expression of NPFF receptors (type 1 and type 2) has been shown in tuberoinfundibular dopaminergic cells and in the ventral tegmental area, which is the site of origin of the mesolimbic dopamine pathway. This evidence supports the hypothesis that NPFF may functionally modulate dopamine-mediated brain functions. The above data have been added to the text (lines 387-401).

  1. This review seems to pretty much focus on the central effect of RFamide, however peripheral stress response could also be affected by circulation RFamides. I would appreciate if authors could make brief comments on their peripheral role.

Thank you for this valuable comment. The focus of the review is indeed on the central effects of RFamide peptides, which was not evident from the title and introduction. This has now been clarified by modifying the title and incorporating this information into the introduction (lines 2, 97, 105, 114-115). Additionally, we have included a brief paragraph on this subject in the Summary (lines 1065-72).

Reviewer 2 Report

Comments and Suggestions for Authors

Kovács et al. review the studies about five groups of the RFamide peptide family by focusing on stress and its related psychpathology.  At first, the basic organization of stress response to homeostatic stressors is schematically shown in Fig. 1.  Then, Table 1 represents the effects of the RFamide peptide family on various physiological phenomena; Figs. 2 and 3 give information about receptors activated by the peptide family and receptors’ distribution in the brain, respectively. Table 2 gives coexpression profile of RFamide peptide-producing neurons.  In the main text, the contents shown in those Figures and Tables are explained for each of the RFamide peptides.  Last, relationship between the RFamide peptides (HPA and SAM) and stress-related disorders (anxiety and depression) is summarized in Fig. 4.  This review article appears to be well organized, but there are many points and comments that may help improve this manuscript, as follows: 

1.     Lines 4 and 5: the numbers should be superscript.

2.     Line 17: not “QFRP” but “QRFP”?

3.     Line 21: “in” following “including” will be unnecessary.

4.     Line 47: please define “CORT”.

5.     Lines 57, 72 and 76: not “AMY” but “Amy” ? (see Fig. 1).

6.     Line 78: “HTH” cannot be found in Fig. 1.  Please amend this point.

7.     Lines 135 and 1111: not “pyroglutamilated” but “pyroglutamylated”?  Not “QFRP” but “QRFP”?  Please check these points.

8.     Line 152: not “FR9a” but “FR9, a”?  Please check this point.

9.     Line 296: please use “IML” (see Fig. 1 and line 1076).

10.  Line 355: please use “HTH”.

11.  It would be better to make the letters in Fig. 3 larger.

12.  Line 411: please put comma following “Since”.

13.  Line 416: please explain “Ki”; please use superscript in Ca2+.

14.  Line 436: please give an explanation for “Takayanagi, 2021 #686”.

15.  Line 462: is “presubiculum subiculum” OK?  Please check this point.

16.  Line 472: not “QRFP receptor” but “QRFPR”?  Please check this point.

17.  Line 485: not “orexin B” but “orexin-B”.

18.  Line 487: not “.. GPCRs form ..” but “.. GPCRs, form ..”.

19.  Line 531: please use superscript in Ca2+.

20.  Line 539: please move “(CXCR4)” behind “receptor”.

21.  Line 546: not “week” but “weak”?  Please check English.

22.  Line 548: please put “gamma”.

23.  Line 553: is “.. was signal was ..” OK?  Please check English.

24.  Line 566: “axis” will be unnecessary.

25.  Line 596: not “RNA-scope” but “RNAscope”? (see line 358)

26.  Line 627: it is not necessary to repeatedly define “PTSD”. (see line 89)

27.  Line 634: does “neurokinin B receptor” mean “NK3”?

28.  Line 647: “axis” will be unnecessary.

29.  Line 671: not “ORFP” but “QRFP”?  Please check this point.

30.  Line 673: please use commas in this line.

31.  Line 681: please put comma following “manner”.

32.  Lines 689 and 691: “beta” should be “β”. (see Table 2)

33.  Line 781: not “metisergide” but methysergide”?  Please check this point.

34.  Line 929: please use subscript in GABAA.

35.  Line 934: not “.. possible the contribution ..” but “.. possible contribution ..”?  Please check English.

36.  Line 976: not “.. axis supported ..” but “.. axis was supported ..”?  Please check English.

37.  Line 977: not “.. evidence. Various ..” but “.. evidence, various ..”?  Please check English.

38.  Line 997: not “.. questions the ..” but “.. questions about the ..”?  Please check English.

39.  Line 1021: not “.. results a ..” but “.. results in a ..”?  Please check English.

40.  The legend of Fig. 4: this makes no mention of “SAM” and therefore should be revised.

41.  Line 1072: please add “pituitary” (see line 244).

42.  There may be more mistakes than pointed out above.  Please check the manuscript once again. 

A large number of studies have been referenced, as can be seen from the 349 citations.  Once again, the authors should check for themselves whether the references are cited correctly.

Comments on the Quality of English Language

The Quality of English Language is good.

Author Response

Thanks to the reviewer for the thorough and accurate correction of MS. We apologize for all minor errors. Below are point by point answers to the problems raised.

Lines 4 and 5: the numbers should be superscript.         Done.

  1. Line 17: not “QFRP” but “QRFP”? Corrected.
  2. Line 21: “in” following “including” will be unnecessary. We deleted it.
  3. Line 47: please define “CORT”. Done.
  4. Lines 57, 72 and 76: not “AMY” but “Amy” ? (see Fig. 1). The correct abbreviation is AMY, the figure, and the legend (the new line number is: 58) has been corrected.
  5. Line 78: “HTH” cannot be found in Fig. 1.  Please amend this point. We amended it (line 79).
  6. Lines 135 and 1111: not “pyroglutamilated” but “pyroglutamylated”?  Not “QFRP” but “QRFP”?  Please check these points. Sorry for the spelling mistakes. They have been corrected (lines 137, 1152).
  7. Line 152: not “FR9a” but “FR9, a”?  Please check this point. Done (line 154).
  8. Line 296: please use “IML” (see Fig. 1 and line 1076). Done (line 298).
  9. Line 355: please use “HTH”. Corrected (line 357).
  10. It would be better to make the letters in Fig. 3 larger. The journal specified the magnification of the figure, which is smaller than the original. We now enlarged the letters in the drawing by one font size, and in the legends by two font sizes. Within the drawing, the size of the characters is limited by space, but the color codes are also very helpful in identifying the labels.
  11. Line 411: please put comma following “Since”. Done (line 426).
  12. Line 416: please explain “Ki”; please use superscript in Ca2+. These have been corrected (line 431).
  13. Line 436: please give an explanation for “Takayanagi, 2021 #686”. Thank you for noticing this error. This was due to a problem with the Endnote program. The article is now cited as article 193, see line 437.
  14. Line 462: is “presubiculum subiculum” OK?  Please check this point. A comma was missing after presubiculum. This has been corrected (line 481).
  15. Line 472: not “QRFP receptor” but “QRFPR”?  Please check this point. It is QRFPR, it has been corrected (line 491).
  16. Line 485: not “orexin B” but “orexin-B”. It has been corrected (line504).
  17. Line 487: not “.. GPCRs form ..” but “.. GPCRs, form ..”. The comma has been added (line 506).
  18. Line 531: please use superscript in Ca2+. Done (line 553).
  19. Line 539: please move “(CXCR4)” behind “receptor”. Done (line561).
  20. Line 546: not “week” but “weak”?  Please check English. It is corrected, thank you.
  21. Line 548: please put “gamma. It is probably a formatting error and has been corrected. Thank you (line 567).
  22. Line 553: is “.. was signal was ..” OK?  Please check English. The first “was” has been deleted (line 572).
  23. Line 566: “axis” will be unnecessary. It has been deleted (line 597) and also throughout the text. The same has been corrected for the HPG axis (line 246).
  24. Line 596: not “RNA-scope” but “RNAscope”? (see line 358) RNAscope is the correct version, thank you (line 627).
  25. Line 627: it is not necessary to repeatedly define “PTSD”. (see line 89) We have deleted the definition (line 658).
  26. Line 634: does “neurokinin B receptor” mean “NK3”? According to the cited reference, both NK1 and NK3 receptors mediate the modulation of dopaminergic, serotoninergic and NA systems. The NK3R is involved in cocaine responses and the NK1R is thought to play a role in the reinstatement of cocaine seeking after extinction. The text has been corrected for clarity. It reads: “However, the NK3 receptor (the cognate receptor of NKB) as well as the NK1 receptor have been shown to mediate the modulation of the dopaminergic, serotoninergic, and NA sys-tems, which are affected by stress-related pathologies. Additionally, the NK3 receptor ap-pears to influence the effects of cocaine, a psychostimulant drug [268].” (lines 663-668,1131).

Lines 1863-64: 268: Jesse R. Schank. Neurokinin Receptors in Drug and Alcohol Addiction. Brain Res. 2020 May 1; 1734: 146729.doi: 10.1016/j.brainres.2020.146729.

  1. Line 647: “axis” will be unnecessary. It has been deleted (line 678).
  2. Line 671: not “ORFP” but “QRFP”?  Please check this point. The spelling error has been corrected (line 702).
  3. Line 673: please use commas in this line. Commas have been added to the sentence (lines 703-706).
  4. Line 681: please put comma following “manner”. Done (line 712).
  5. Lines 689 and 691: “beta” should be “β”. (see Table 2). It has been changed (lines 720, 721).
  6. Line 781: not “metisergide” but “methysergide”?  Please check this point. We have corrected this, thank you (line 812).
  7. Line 929: please use subscript in GABAA. Done (line 960).
  8. Line 934: not “.. possible the contribution ..” but “.. possible contribution ..”?  Please check English. It has been corrected (line 965).
  9. Line 976: not “.. axis supported ..” but “.. axis was supported ..”?  Please check English. The text has been corrected accordingly (line 1007).
  10. Line 977: not “.. evidence. Various ..” but “.. evidence, various ..”?  Please check English. We tried to avoid very long sentences, so we use two sentences here instead of one. To make the text clearer, we added "For example" before the second sentence. It reads: “Although the exact mechanisms of the effect of KP in stress remain to be clarified, the negative impact of stress on the KP/Kiss1R system and the reproductive axis supported by a substantial body of evidence. For example, various types of stressors, including lipopolysaccharide-induced inflammation, acute restraint or insulin-induced hypoglycemia, disrupted LH pulsatility, decreased KP expression or KP neuron activity, and altered Kiss1R mRNA expression in the HTH of female rodents” (lines 1006-1011).
  11. Line 997: not “.. questions the ..” but “.. questions about the ..”?  Please check English. The text has been corrected accordingly (line 1029).
  12. Line 1021: not “.. results a ..” but “.. results in a ..”?  Please check English. The text has been corrected accordingly (line 1053).
  13. The legend of Fig. 4: this makes no mention of “SAM” and therefore should be revised. The legend has been corrected according to the suggestion (line 1042). Thank you.
  14. Line 1072: please add “pituitary” (see line 244). Done (line 1112).
  15. There may be more mistakes than pointed out above.  Please check the manuscript once again.  We have checked the text again (line 1001).

A large number of studies have been referenced, as can be seen from the 349 citations.  Once again, the authors should check for themselves whether the references are cited correctly.

Thank you for your suggestion, we have checked the citations.